# The Price of Opportunity Fairness in Matroid Allocation Problems

**Rémi Castera**[*]
Moroccan Center for Game Theory
University Mohammed VI Polytechnic
Rabat, Morocco

**Felipe Garrido-Lucero**
IRIT, Université Toulouse Capitole
Toulouse, France

**Patrick Loiseau**
Inria, Fairplay joint team
Palaiseau, France

**Simon Mauras**
Inria, Fairplay joint team
Palaiseau, France

**Mathieu Molina**
Inria, Fairplay joint team
Crest, ENSAE
Palaiseau, France

**Vianney Perchet**
ENSAE, Fairplay joint team
Criteo AI Lab
Palaiseau, France

## Abstract

We consider matroid allocation problems under *opportunity fairness* constraints: resources need to be allocated to a set of agents under matroid constraints (which include classical problems such as bipartite matching). Agents are divided into $C$ groups according to a sensitive attribute, and an allocation is opportunity-fair if each group receives the same share proportional to the maximum feasible allocation it could achieve in isolation. We study the Price of Fairness (PoF), i.e., the ratio between maximum size allocations and maximum size opportunity-fair allocations. We first provide a characterization of the PoF leveraging the underlying polymatroid structure of the allocation problem. Based on this characterization, we prove bounds on the PoF in various settings from fully adversarial (worst-case) to fully random. Notably, one of our main results considers an arbitrary matroid structure with agents randomly divided into groups. In this setting, we prove a PoF bound as a function of the (relative) size of the largest group. Our result implies that, as long as there is no dominant group (i.e., the largest group is not too large), opportunity fairness constraints do not induce any loss of social welfare (defined as the allocation size). Overall, our results give insights into which aspects of the problem's structure affect the trade-off between opportunity fairness and social welfare.

## 1 Introduction

Allocating scarce resources among agents is a fundamental task in diverse fields such as online markets [20], online advertising [54], the labor market [22], university admissions [3, 38], refugee programs [1, 26, 36], or organ transplants [2]. Traditionally, central planners aim to efficiently compute allocations that maximize some metric of social welfare such as the total number of allocated resources. Unfortunately, optimal allocations that neglect equity considerations often result in disparate treatment and unfair outcomes for legally protected groups of individuals, as documented in domains such as job offerings [50, 63] and online advertising [5, 14], among others.

Matching markets, a prominent instance of resource allocation problems, are especially sensitive to these discrimination concerns. For example, initiatives such as the European Union's proposed

---

[*]This research was partly conducted while at Université Grenoble Alpes, Inria, CNRS, Grenoble INP, LIG.

job-matching platform for migrants [22], and the urgent demands arising from the global refugee crisis [34] are complex challenges where fairness must be accounted for: migrants can belong to different demographic groups defined by sensitive attributes such as age, ethnicity, gender, or wealth, and jobs or resettlement locations must be allocated in a fair manner.

Motivated by these challenges, we consider *matroid allocation problems*, i.e., resource allocation problems where the constraint has a matroid structure [61, Chapter 44]. An instance of our problem is defined by a pair $(E, \mathcal{I})$ where $E$ is a finite set of agents and $\mathcal{I}$ the set of feasible resource allocations, and such that $E$ is partitioned based on sensitive attributes into $C \in \mathbb{N}$ distinct groups (for the formal definitions, please refer to Section 2). Each feasible solution selects a subset of agents who can simultaneously receive a resource, with each agent receiving at most one. Matroid allocation problems have a theoretical structure that gives tractability while being expressive enough to formulate many important problems. They include our main application of bipartite matching discussed above, but also other allocation problems where fairness is relevant. For instance, the allocation of research funding within a university, where the goal is to fairly distribute limited departmental budgets among professors while respecting diversity criteria such as seniority or gender balance; or the assignment of faculty positions across departments and teams, where office space and departmental capacities must be respected; can both be formulated as matroid allocation problems corresponding respectively to partition and laminar matroids. From a combinatorial optimization perspective, this constitutes a *selection problem* under matroid constraints. We use the term "matroid allocation" to highlight the fairness objective, as we focus on how resources are effectively allocated across demographic groups.

Prior works have tackled fairness challenges in (matroid) allocation problems. Chierichetti et al. [18] study fair matroid allocation problems, and Bandyapadhyay et al. [9] examine fair matching (see a more complete discussion in Appendix A). These works, however, focus on the efficient computation of fair allocations and provide approximation algorithms. In contrast, we focus on the *quality* of the optimal fair allocation. Indeed, imposing fairness constraints may reduce *social welfare* (the total number of resources allocated). To understand the trade-off between fairness and social welfare, we study the metric called *Price of Fairness*, introduced independently in [11] and [15] in the contexts of proportional fairness and equitability, respectively. This metric, formally defined by

$$\mathrm{PoF}(\mathcal{I}) := \frac{\max\{|S| : S \in \mathcal{I}\}}{\max\{|S| : S \in \mathcal{I} \text{ is fair}\}},$$

where $|S|$ denotes the size of the allocation $S$, provides insights into the scenarios where fairness leads to a degradation of the social welfare.

We focus on a novel notion of fairness that we introduce: **opportunity fairness**. Opportunity fairness draws inspiration from the notion of Equality of Opportunity [42] in machine learning and from Kalai-Smorodinsky fairness [47] in fair division. This fairness notion is particularly adapted to the structure of the allocation problem as it accounts for the inherent capabilities of the agents groups. Formally, an allocation is called opportunity fair if for any two groups $c, c'$, it satisfies

$$\frac{\text{\# of resources allocated to } c}{\text{Total \# of resources that can be allocated to } c} = \frac{\text{\# of resources allocated to } c'}{\text{Total \# of resources that can be allocated to } c'}.$$

We consider a **large market setting** where the number of resources that can be allocated to each group is large, as is often the case, e.g., in job market platforms. In this situation, integral allocations can be well approximated by fractional—or randomized—allocations (as proved in Section 2.3). Hence, we focus on the PoF computation under *fractional allocations*.

**Contributions**. The price of opportunity fairness may depend on different features of the problem, in particular the set of feasible allocation and the agents' group assignment. We prove tight PoF bounds in multiple settings from adversarial to fully random, in line with the beyond-worst-case paradigm [59], which seeks to capture more realistic and nuanced behaviors than worst-case analysis alone:

1. *Polymatroid representation:* We first show that for matroids constraints, the set of feasible per-group allocations can be represented as a *polymatroid*, a multi-set generalization of matroids. We then leverage the polymatroid representation to achieve a simpler characterization for the price of opportunity fairness, which is key for the subsequent analysis (Proposition 3.2).

2. *Adversarial analysis:* When both the group partition and the set of feasible allocations are chosen adversarially, we show that the worst-case PoF is $C - 1$, regardless of the number of agents (Theorem 4.1).

3. *Parametrized families of matroids:* By considering a parametrized family of matroids, we conduct a finer PoF analysis with bounds that interpolate the best case of no loss, $\mathrm{PoF} = 1$, and the worst possible case, $\mathrm{PoF} = C - 1$ (Propositions 4.4 and 4.6).

4. *Semi-random setting:* When agents are randomly partitioned into $C$ groups according to some distribution $p = (p_1, \cdots, p_C)$, we characterize the worst-case PoF as a function of $\max_{c \in [C]} p_c$ by reducing a joint infinite-dimensional combinatorial optimization problem to a one-dimensional optimization problem (Theorem 4.8). Remarkably, we show that as long as $\max_{c \in [C]} p_c \leq 1/(C - 1)$, no social welfare loss is incurred under opportunity fairness constraints, in particular suggesting that *the trade-off between fairness and social welfare is not entirely due to the presence of small groups, but rather due to the presence of a dominant group* (Corollary 4.9).

5. *Random graphs model:* Finally, we extend the no-social-welfare-loss result of the previous case to any groups partition distribution $p$ whenever the set of feasible allocations $\mathcal{I}$ is obtained from certain Erdös-Rényi random graphs (Propositions 4.10 and 4.11).

Overall, our results lead to a better understanding of how the structure of both the agents groups and feasible allocations can affect the price of fairness. The main qualitative takeaway is that *for realistic matroid and protected groups instances, opportunity fair allocations incur only a small social welfare loss.* While our main focus is on opportunity fairness, as it is the most relevant fairness notion for the setting we consider, we also provide additional results for other fairness notions in Appendix F.

## 2 Model

### 2.1 Matroids and Colored Matroids

Let $E$ be a finite set of agents. We denote by $\mathcal{I} \subseteq 2^E$ a family of **feasible allocations**, where for any allocation $S \in \mathcal{I}$, $e \in S$ represents that agent $e$ got a resource allocated. We assume that $(E, \mathcal{I})$ is a finite matroid:

**Definition 2.1.** The pair $(E, \mathcal{I})$ is a (finite) **matroid** if $\emptyset \in \mathcal{I}$ and the following properties are satisfied: (i) For any $S \in \mathcal{I}$ and $T \subseteq S$, $T \in \mathcal{I}$ (*hereditary property*); and (ii) For any $S, T \in \mathcal{I}$, such that $|S| < |T|$, there exists $e \in T \setminus S$ such that $S \cup \{e\} \in \mathcal{I}$ (*augmentation property*).

Matroids are particularly useful for combinatorial optimization. They are rich enough to describe many allocation problems often encountered in practice, e.g., the bipartite matching (traversal matroids), the allocation of research funding within a university (partition matroids), or the assignment of faculty positions across departments and teams (laminar matroids), mentioned in the introduction (see Appendix B for the definition of each sub-class of matroids). More importantly, due to the augmentation property, a maximal size allocation under matroid constraints can be computed in polynomial time via a greedy algorithm, provided there is a polynomial-time oracle to identify if a set is feasible.

To every matroid $(E, \mathcal{I})$, we associate a **rank function** $\mathrm{r} : 2^E \to \mathbb{R}_+$, which maps each $S \subseteq E$ to $\mathrm{r}(S) := \max\{|T| \mid T \subseteq S, T \in \mathcal{I}\}$, that is, to the size of the maximum feasible allocation included in $S$. Basic results of matroid theory [61] show that the rank function is submodular[2], non-decreasing, and that $0 \leq \mathrm{r}(S) \leq |S|$ for any $S \subseteq E$.

We consider matroids in which the set of agents is partitioned into $C$ groups (based on sensitive attributes)—also termed colors [18]. Given a matroid $(E, \mathcal{I})$, $C \in \mathbb{N}$, and $(E_c)_{c \in [C]}$ a partition of $E$ into groups, the tuple $((E_c)_{c \in [C]}, \mathcal{I})$ is called a ***C*-colored matroid** (or simply colored matroid).

Given a colored matroid and a subset of groups $\Lambda \subseteq [C]$, we denote by $\mathrm{r}(\Lambda) := \mathrm{r}(\cup_{c \in \Lambda} E_c)$ the rank of the corresponding subset of agents. We call the function $\mathrm{r} : 2^{[C]} \to \mathbb{R}_+$ the rank function of the colored matroid. The rank function of the colored matroid inherits all properties from the rank function of the original matroid. In addition, remark that $\mathrm{r}([C])$ corresponds to the size of a maximum size allocation within $\mathcal{I}$, i.e., the maximum social welfare achievable in the corresponding resource allocation problem, while $\mathrm{r}(c)$[3] corresponds to the *opportunity level* of the color (the group) $c$, i.e., the maximum social welfare when considering only the agents within $E_c$.

---

[2] A set function $f$ is submodular if for all finite sets $S$ and $T$, $f(S) + f(T) \geq f(S \cap T) + f(S \cup T)$.

[3] We write $\mathrm{r}(c)$ instead of $\mathrm{r}(\{c\})$ for convenience, since there is no ambiguity.

## 2.2 Opportunity Fairness

Given a colored matroid $((E_c)_{c \in [C]}, \mathcal{I})$, we denote $\mathrm{I} := \{x \in \mathbb{N}^C \mid$ there exists $S \in \mathcal{I}, x_c = |S \cap E_c|,$ for any $c \in [C]\}$. The set $\mathrm{I}$ of **integer feasible group allocations** will be sufficient to find optimal fair allocations. We consider fractional allocations as feasible solutions (see Section 2.3 for the justification): we denote by $M := \text{convex hull}(\mathrm{I})$ the set of **fractional feasible group allocations**.

With this notation, we introduce opportunity fairness and the price of fairness for fractional allocations:

**Definition 2.2.** A fractional allocation $x \in M$ is **opportunity fair** if for any $c, c' \in [C]$,

$$\frac{x_c}{\mathrm{r}(c)} = \frac{x_{c'}}{\mathrm{r}(c')}.$$

We denote by $F$ the **set of opportunity fair fractional allocations**. The idea behind this notion is that each group is entitled to a quantity of resources proportional to the maximal quantity this group can get across all feasible allocations.

**Example.** Consider a refugee resettlement task with $n$ men and $m$ women, each linked to cities in a bipartite graph. At most $r_1$ men can be matched, and $r_2$ women can be matched. Let the size of the largest allocation be $r_{1,2} \leq r_1 + r_2$. Under the optimal opportunity fair matching, each group's share is proportional to its isolated capacity, relocating $r_1 r_{1,2}/(r_1 + r_2)$ men and $r_2 r_{1,2}/(r_1 + r_2)$ women.

The social welfare of an allocation $x$, is simply the total number of resources allocated, that is to say $\sum_{c \in [C]} x_c$. Then the **Price of Opportunity Fairness** (PoF) is defined as

$$\text{PoF}(M) := \frac{\max_{x \in M} \sum_{c \in [C]} x_c}{\max_{x \in F} \sum_{c \in [C]} x_c}.$$

This quantity is always greater than 1 and grows as the fairness constraint becomes more restrictive. In the previous example, the Price of Opportunity Fairness equals 1, indicating no loss in total utility.

## 2.3 From Fractional Allocations to Approximately Fair Integral Allocations

While fractional allocations are not directly implementable, a natural and practical interpretation of a fractional allocation $x$ is as a randomized integral allocation. Indeed, since $M$ is a convex polytope, every point in $M$ can be expressed as a convex combination of its vertices. Moreover, as $M$ is a polymatroid defined by an integral submodular function, all its vertices are integral [30]. Consequently, the convex coefficient associated with each vertex can be interpreted as the probability mass assigned to the corresponding integral allocation.

We consider the relaxation to fractional allocations for two main reasons. First, when restricted to integer allocations, many resource allocation problems have no opportunity fair allocations besides the empty one. Indeed, even for two colors, whenever $\mathrm{r}(1)$ and $\mathrm{r}(2)$ are co-prime and $(\mathrm{r}(1), \mathrm{r}(2)) \notin \mathrm{I}$, the only feasible fair integral allocation is to allocate 0 resources to each group (see Appendix C).

Second, by slightly relaxing the fairness constraints, any optimal fair fractional allocation can be implemented in an integral fashion through a specific randomized rounding technique, at a cost that becomes negligibly small in large markets. We first define the relaxed fairness notion:

**Definition 2.3.** For $\gamma \in [0, 1]$, an allocation $x \in M$ is said to be **$\gamma$-opportunity fair** if for any $c, c' \in [C]$, it holds that $x_c/\mathrm{r}(c) \geq \gamma \cdot x_{c'}/\mathrm{r}(c')$.

Setting $\gamma = 1$ recovers Definition 2.2, while $\gamma = 0$ corresponds to no fairness considerations.

**Proposition 2.4.** *For any fractional opportunity fair maximum size allocation $x \in F$, there exists a random allocation $X$, such that, $\mathbb{E}[X] = x$ and for all realizations, $X$ is feasible, integral, $\left(1 - \frac{2C}{\min_c \mathrm{r}(c)}\right)$-opportunity fair, and $\|X - x\|_1 \leq C$. Conversely, these bounds are tight up to constants: there exist instances $(M, x)$ and $(M', x')$ such that any rounding scheme $X$ is at most $1 - O(\frac{C}{\min_c \mathrm{r}(c)})$-opportunity fair in the first case, or satisfies $\|X - x'\|_1 \geq \Omega(C)$ in the second case.*

*Proof Sketch.* The argument relies on the polymatroid characterization of $M$ (Proposition 3.2) that will be proven independently in Section 3. Then by Theorem 35 of [30], which guarantees that the extreme points of the intersection of two integral polymatroids remain integral, $x$ can be written as a convex combination of nearby integral allocations. See Appendix E.1 for the proof. $\qquad\square$

We remark that the distribution of $X$ can be computed in polynomial time by using an algorithmic version of Carathéodory's Theorem (see Theorem 6.5.11 of [40]). In a **large market**, that is, when $\min_{c \in [C]} r(c)$ is large, Proposition 2.4 shows that both the gap between $\text{PoF} = r([C])/\|x\|_1$ and $r([C])/\|X\|_1$, and the fairness degradation, become negligible. Hence, it provides the strongest fairness guarantees ex-ante (since $\mathbb{E}[X] = x$ and $x$ is perfectly fair), jointly with an approximately fair integral allocation at low cost ex-post. Such a best-of-both-worlds approach that leverages fractional allocations to design lotteries satisfying ex-ante requirements (here fairness) and additional ex-post properties is widespread in market design [4, 12, 13, 46] and fair division [8, 35], with applications in school choice, housing, and kidney exchange, to name but a few.

Even when considering fractional allocations, one may want to impose approximate fairness constraints only. Interestingly, for any colored matroid, our bounds on PoF (with exact fairness constraint from Definition 2.2) can easily be transposed into bounds for the relaxed setting. Formally, denote by $\text{PoF}_\gamma$ the price of $\gamma$-opportunity fairness (i.e., when fairness is defined by Definition 2.3).

**Proposition 2.5.** *Let $M$ be a $C$-colored matroid. Then, either $\gamma < \max_{x \in F} \min_{c \in [C]} x_c / r(c)$ and $\text{PoF}_\gamma(M) = 1$, or*

$$\gamma \cdot \text{PoF}(M) \leq \text{PoF}_\gamma(M) \leq \frac{\gamma \cdot \text{PoF}(M)}{1 - (1 - \gamma)^{\frac{((C-1) - \text{PoF}(M))}{C-2}}}.$$

See proof in Appendix E.2. Proposition 2.5 allows us to seamlessly lift any PoF upper and lower bounds to $\text{PoF}_\gamma$, with $\text{PoF}_\gamma \approx \gamma \text{PoF}$ for small $\gamma$. As before, the same randomized rounding applies, at a small cost. Thus, **the rest of the paper will focus on fractional allocation and perfect fairness constraints**, as Propositions 2.4 and 2.5 can handle integrality and relaxed constraint considerations.

## 2.4 Comparison with proportional fairness and other fairness notions

*Proportional fairness*, a broadly studied fairness notion in the literature, aims to equalize the ratios $x_c / |E_c|$, i.e., the number of allocated resources to each group relative to their size. In machine learning, proportional fairness corresponds to *demographic parity* [10] while *opportunity fairness* is more closely related to *equality of opportunity*, as it accounts for the inherent quality of the groups.

A key limitation of proportional fairness in the matroid allocation setting is that it is *sensitive to the presence of irrelevant agents*, unlike opportunity fairness. Indeed, whenever the allocation problem possesses agents that cannot be allocated any resources, proportional fairness becomes too constraining to satisfy. For instance, adding irrelevant agents of color $c$ increases $|E_c|$ without increasing the size of the feasible allocations to that group, thereby reducing the ratio $x_c / |E_c|$. To maintain proportional fairness, the allocation to other groups must be reduced, even though the underlying allocation problem remains unchanged.

This highlights a pathological behavior: even under fractional allocations, the *Price of Proportional Fairness* can be unbounded, as adding irrelevant agents drives all fair allocations for other groups to zero. In contrast, as we show throughout the paper, *opportunity fairness is robust to such manipulations* and remains bounded, since it accounts for the structure of the allocation problem. In Appendix F, we further discuss the relationships between different notions of fairness (including weighted and leximin fairness) and their respective prices of fairness, underscoring the relevance of opportunity fairness in our context.

## 3 Polymatroid Structure and PoF Characterization

This section is devoted to characterizing the price of opportunity fairness of a matroid as a simple combinatorial optimization problem. Our main technique will be the use of polymatroids.

**Definition 3.1.** The **polymatroid** associated to the submodular function $f : 2^C \to \mathbb{R}_+$ is the polytope

$$\left\{ x \in \mathbb{R}_+^C \mid \sum_{c \in \Lambda} x_c \leq f(\Lambda), \forall \Lambda \subseteq [C] \right\}.$$

Polymatroids can be seen as a generalization of matroids, as there is a natural mapping from a matroid on a ground set $E$ to a polymatroid included in $[0, 1]^E$, where a feasible allocation $S \in \mathcal{I}$ is associated to a vector $z \in [0, 1]^E$ with coordinates $z_e = 1$ if $e \in S$ and 0 otherwise. Polymatroids are strictly more general, as coordinates can be larger than 1. Proposition 3.2 shows that there is also a natural relation between colored matroids and polymatroids. Refer to Appendix E.3 for the proof.

**Proposition 3.2.** *Let* $((E_c)_{c\in[C]}, \mathcal{I})$ *be a colored matroid with rank function* r *and set of feasible fractional allocations* $M$. *Then,* $M$ *is the polymatroid associated to the function* r, *i.e.,*

$$M = \big\{ x \in \mathbb{R}_+^C \mid \sum\nolimits_{c\in\Lambda} x_c \leq \mathrm{r}(\Lambda), \forall \Lambda \subseteq [C] \big\}.$$

Note that while the usual natural mapping from matroids to polymatroids is not a surjection (the coordinates must remain bounded by 1, which is not the case of all polymatroids), the mapping $((E_c)_{c\in[C]}, \mathcal{I}) \mapsto M$ from the set of colored matroids to the set of all integral Polymatroids is a surjection. From now on, we will use interchangeably the names feasible fractional allocations, polymatroid, and colored matroid for $M$.

The set $M$ inherits interesting properties from being a polymatroid. For instance, the Pareto frontier of the Multi-objective Optimization Problem $\max_{x\in M}(x_1, x_2, ..., x_C)$ corresponds to the set of allocations maximizing social welfare $\sum_{c\in[C]} x_c$ [43]. In particular, the existence of an allocation of maximum size that is opportunity fair is reduced to verifying whether the intersection between the Pareto frontier and the line defined by the opportunity fairness condition is non-empty. Figure 1 illustrates $M$ and $P$ for a $C$-colored matroid with $C = 2$ and $C = 3$, respectively. Remark that $P$ is simultaneously the Pareto frontier and the set of points which maximize $\sum_{c\in[C]} x_c$.

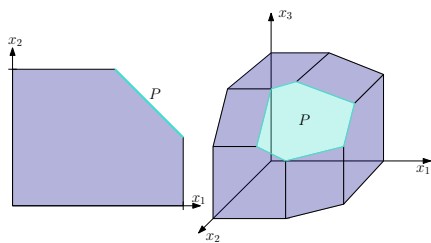

Figure 1: Examples of the set of fractional feasible allocations $M$ (dark blue solid region) and the Pareto frontier $P$ (light blue region) for $C = 2$ and $C = 3$.

The structure of $M$ yields the following characterization of PoF. Refer to Appendix E.4 for the proof.

**Corollary 3.3.** *The price of opportunity fairness of a polymatroid* $M$ *is given by,*

$$\mathrm{PoF}(M) = \frac{\mathrm{r}([C])}{\sum_{c\in[C]} \mathrm{r}(c)} \cdot \max_{\Lambda \subseteq [C]} \frac{\sum_{c\in\Lambda} \mathrm{r}(c)}{\mathrm{r}(\Lambda)}. \tag{1}$$

Remark that the combinatorial optimization problem in (1) is exponential in $C$. Even though real-life applications typically involve a small number of sensitive attributes (often only 2), applications involving intersectional fairness between different sensitive features may introduce a larger number of colors [14, 49, 56]. The PoF can be computed in time $\mathrm{poly}(C, |E|)$ whenever the underlying matroid possesses a **polynomial-time independence oracle** as is the case of transversal, graphic, partition, laminar and uniform matroids. Refer to Appendix D for the details.

In the following section we leverage Corollary 3.3 to tightly bound the PoF in various settings.

## 4 Bounding the Price of Fairness

### 4.1 Adversarial Price of Fairness

Our first result is to show that the price of opportunity fairness is always bounded, with a bound only depending linearly on $C$, the number of colors of the colored matroid, independent of the number of agents $|E|$ and the number of feasible allocations $|\mathcal{I}|$. Refer to Appendix E.5 for the proof.

**Theorem 4.1.** *For any* $C$-*colored matroid* $M$, *we have* $\mathrm{PoF}(M) \leq C - 1$, *and this bound is tight.*

Theorem 4.1 implies the following remarkable result.

**Corollary 4.2.** *For any* 2-*colored matroid* $M$, $\mathrm{PoF}(M) = 1$.

Corollary 4.2 shows that whenever agents are divided in two groups, no social welfare loss is incurred due to the opportunity fairness constraint. For an alternative geometrical proof of Corollary 4.2, please refer to Appendix E.6. As a direct application of Proposition 2.5 and Theorem 4.1, we obtain the following corollary, which follows immediately from the monotonicity of the upper bound in Proposition 2.5 with respect to $\mathrm{PoF}(M)$.

**Corollary 4.3.** *Let* $\gamma \in [0, 1]$. *For any* $C$-*colored matroid* $M$, *we have* $\mathrm{PoF}_\gamma(M) \leq \gamma \cdot (C - 1)$, *and this bound is tight.*

Theorem 4.1 raises the question of whether tighter bounds can be obtained by restricting the resource allocation problem to specific subclasses of matroids. However, the bound is in fact tight for any class of matroids that contains either graphic or partition matroids, which implies tightness for transversal and laminar matroids. Refer to Appendix E.5 for the details.

## 4.2 Parametric Price of Fairness

The worst-case bound derived in the previous section relies on the existence of a specific polymatroid, as outlined in the proof of Theorem 4.1. Essentially, this requires an underlying structure where one group $E_c$ can have a rank that grows arbitrarily large while the ranks of other groups remain bounded. This raises the question of whether more favorable guarantees for the price of opportunity fairness can be attained when all groups exhibit similar ranks. Proposition 4.4 provides an upper bound on PoF based on the ranks of the groups. The proof is detailed in Appendix E.7.

**Proposition 4.4.** *For any $C$-colored matroid $M$, it holds,*

$$\text{PoF}(M) \leq \frac{1}{2} \cdot \frac{\max_{c \in [C]} \text{r}(c)}{\min_{c \in [C]} \text{r}(c)} + \frac{C}{4} \cdot \left( \frac{\max_{c \in [C]} \text{r}(c)}{\min_{c \in [C]} \text{r}(c)} \right)^2 + \frac{1}{4C} \cdot \mathbb{1}\{C \text{ odd}\}.$$

*Moreover, whenever all groups have the same rank, the resulting bound is tight.*

The bound in Proposition 4.4 takes into account the shape of the polytope $M$. When all colors have the same rank, PoF scales as $C/4$. While this upper bound is smaller than the one stated in Theorem 4.1, the price of fairness remains linear with respect to the number of colors.

To complement the analysis, we consider another geometrical parameter related to the shape of $M$ that can interpolate PoF between 1 and $C/4$. Intuitively, PoF is expected to be low when either there is no competition between groups, or the competition is extremely fierce and no group can be unilaterally allocated resources without damaging the allocation of others. Similar behavior has been observed for the Price of Anarchy in congestion games [21], which approximates to 1 under both light and heavy traffic conditions. In the context of the price of opportunity fairness, the relevant problem complexity measure is the associated *independence index* that we define below.

**Definition 4.5.** We define the **independence index** of a polymatroid $M$ as $\rho(M) := \frac{\text{r}([C])}{\sum_{c \in [C]} \text{r}(c)}$.

The independence index measures how close the maximal social welfare $\text{r}([C])$ is to the social welfare of the *utopian allocation*, the allocation where each group $E_c$ receives $\text{r}(c)$ resources (which, in general, is not a feasible allocation). Note that $\rho$ always falls within the interval $[1/C, 1]$, with $\rho = 1/C$ corresponding to complete competition between groups, and $\rho = 1$ corresponding to full independence between groups. These extreme values of the independence index impose a distinct shape on $M$, as illustrated in Figure 3.

**Proposition 4.6.** *Let $M$ be a $C$-colored matroid. Suppose that for any $c, c'$ in $[C]$, $\text{r}(c) = \text{r}(c')$. Whenever $\rho \in [1/C, 1/(C-1)]$, $\text{PoF}(M) = 1$. Otherwise,*

$$\text{PoF}(M) \leq \rho \max \left( \frac{C - \lfloor C\rho \rfloor + 1}{C\rho - \lfloor C\rho \rfloor + 1}, C - \lfloor C\rho \rfloor \right) \leq \rho((1 - \rho)C + 1).$$

*In addition, the first upper bound is jointly tight in $\rho$ and $C$.*

The proof of Proposition 4.6 is provided in Appendix E.8, where Figure 11 illustrates a tight example of the first upper bound for a transversal matroid. Figure 2 illustrates both upper bounds from Proposition 4.6 for $C = 5$ groups with equal ranks. We observe that in both extremes, PoF tends towards 1. Notice that the second upper bound in Proposition 4.6, when maximized over $\rho$, aligns with the bound from Proposition 4.4 (when all groups have identical ranks). Therefore, the independence index interpolates the price of opportunity fairness between 1 and an order of $C/4$ when all groups have the same isolated social welfare.

Worst-case analysis results stand out by their robustness. However, the particular matroid examples attaining the upper bounds (even under the extra structural assumptions) are rarely observed in real-life. Due to this, the following sections will be dedicated to analyzing PoF in random settings.

## 4.3 Semi-Random Price of Fairness (Random Coloring)

Our first random setting considers an adversarial matroid choice with a random group agents partition. Formally, we denote by $\Delta^C$ the simplex of dimension $C$, that is, the set of all vectors $p \in [0, 1]^C$

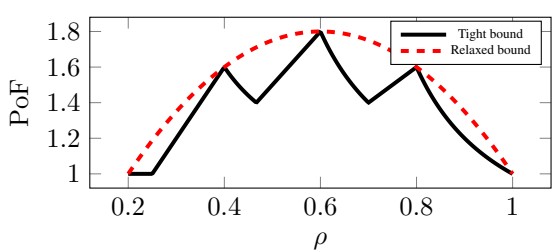

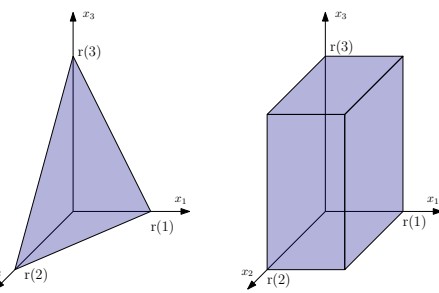

Figure 2: PoF upper bounds stated in Proposition 4.6 for 5 groups with equal rank, with variable value of the independence index $\rho$.

Figure 3: Shape of $M$ for extreme values of independence index. Left: $\rho = 1/C$, right: $\rho = 1$.

such that $\sum_{c \in C} p_c = 1$. Given a vector $p \in \Delta^C$ without null entries, we create a random partition of a matroid $(E, \mathcal{I})$ (a coloring of the elements in $E$) by independently and identically assigning each element $e \in E$ to $c \in [C]$ with probability $p_c$. We denote by $M(p)$ the polymatroid obtained by the random coloring of the agents in $E$ according to the vector $p$.

Let $(M_n(p))_{n \in \mathbb{N}}$ be a sequence of C-colored matroids over sets $(E^{(n)})$ such that $|E^{(n)}| = n$, randomly colored according to $p$, and $(\mathrm{r}_n)_{n \in \mathbb{N}}$ the associated sequence of rank functions. For each $c \in [C]$, suppose the following limit exists,

$$R(p_c) := \lim_{n \to \infty} \frac{\mathbb{E}_{p_c}[\mathrm{r}_n(c)]}{n}, \tag{2}$$

where $\mathbb{E}_{p_c}[\mathrm{r}_n(c)]/n$ represents the rescaled expected social welfare of group $c$. Remark that assuming convergence is not particularly restrictive as $\mathbb{E}_p[\mathrm{r}_n(c)]/n$ is bounded in $[0, 1]$ and thus, it always admits a converging subsequence. We extend the previous definition to any subset $\Lambda \subseteq [C]$ by,

$$R\left(\sum_{c \in \Lambda} p_c\right) := \lim_{n \to \infty} \frac{1}{n} \cdot \mathbb{E}_{(p_c)_{c \in \Lambda}}\left[\mathrm{r}_n(\cup_{c \in \Lambda} E_c)\right].$$

Finally, we will assume that $\mathrm{r}_n([C]) = \Omega(n)$, which ensures that the size of the optimal allocation grows with the size of the ground sets $E^{(n)}$. The semi-random model represents settings where agents' protected attributes are independent of their individual capabilities or opportunities, i.e., the matroid structure. Random coloring therefore isolates the impact of feasibility constraints and group size, relative to the protected attributes, in generating unfairness.

**Remark 1.** The use of a sequence of colored matroids $(M_n(p))_{n \in \mathbb{N}}$ is mainly a technical device to capture the large-scale limit. Conceptually, this can be viewed as selecting a large adversarial matroid (whose rank grows roughly proportionally to the size of its ground set) and then randomly coloring its elements according to $p$.

We first show that the price of opportunity fairness for large colored matroids is completely characterized by the function $R$.

**Proposition 4.7.** *If* $\liminf_{n \to \infty} \frac{r_n([C])}{n} > 0$, *then*

$$\mathrm{PoF}(M_n(p)) \xrightarrow[n \to \infty]{\mathrm{P}} \max_{\Lambda \subseteq [C]} \frac{R(1)}{\sum_{c \in [C]} R(p_c)} \cdot \frac{\sum_{c \in \Lambda} R(p_c)}{R(\sum_{c \in \Lambda} p_c)}, \tag{3}$$

*where* $\xrightarrow{\mathrm{P}}$ *denotes convergence in probability. Moreover, the function $R$ is such that $R(0) = 0$, $R$ is concave, non-decreasing, and 1-Lipschitz. Conversely, any function with these three properties can be realized as the limit of a sequence of limit-matroid derived functions $R'_m$, i.e.* $\|R - R'_m\|_\infty \xrightarrow[m \to \infty]{} 0$.

*Proof sketch.* We first show that, as $R$ is the multi-linear extension of a submodular function, it satisfies the aforementioned properties. Using the concavity of $R$, we show that whenever $\mathrm{r}([C])$ is large, with high probability, $\mathrm{r}(c)$ is large as well. Then, by using McDiarmid's concentration inequality, the convergence in probability is concluded. The approximation result is proved by constructing a family of simple functions from specific sequences $(M_n)_n$ whose closed convex hull is equal to the desired set of functions. The full proof is included in Appendix E.9. □

The first property of Proposition 4.7 shows that upper bounding the right-hand side of Equation (3) yields an upper bound on $\mathrm{PoF}$. The second part shows an equivalence between sequences of $C$-colored matroids and the set of concave, non-decreasing, 1-Lipschitz functions, which we denote by $\mathcal{R}$. Therefore, we can shift the problem of bounding the price of opportunity fairness of $C$-colored matroids to bounding the right-hand side of Equation (3) over all functions in $\mathcal{R}$. We aim to find a bound that depends only on $\max_{c \in [C]} p_c$. The following theorem provides the exact solution:

**Theorem 4.8.** *Fix* $\pi \in [1/C, 1]$ *and consider* $\Delta_\pi^C := \{p \in \Delta^C \mid \max_{c \in [C]} p_c = \pi\}$ *the set of probability distributions with maximum value of* $\pi$. *It follows that*

$$\max_{p \in \Delta_\pi^C} \max_{R \in \mathcal{R}} \max_{\Lambda \subseteq [C]} \frac{R(1)}{\sum_{c \in [C]} R(p_c)} \frac{\sum_{c \in \Lambda} R(p_c)}{R(\sum_{c \in \Lambda} p_c)} = \max_{\lambda \in [C]} \psi_\lambda \left( \frac{1 - (C - \lambda)\pi}{C} \right) \leq \min(C - \frac{1}{\pi}, 1), \quad (4)$$

*where* $\psi_\lambda : [-\lambda, \frac{1}{C}] \to \mathbb{R}$, *for each* $\lambda \in [C]$, *is given by*

$$\psi_\lambda(q) = \begin{cases} \lambda & \text{if } q \in [-\lambda, 0], \\ \frac{\lambda}{(\lambda C q - 1)^2} \cdot \left(1 + q(1-2\lambda) + C(\lambda - 2 + \lambda q)q - 2\sqrt{(\lambda-1)(C-1)(1-Cq)(1-\lambda q)q}\right) & \text{if } q \in \left(0, \frac{(\lambda-1)}{\lambda(C-1)}\right], \\ 1 & \text{if } q \in \left(\frac{(\lambda-1)}{\lambda(C-1)}, \frac{1}{C}\right]. \end{cases}$$

*Proof sketch.* The left-hand side of Equation (4) defines an infinite-dimensional combinatorial optimization problem, for which classical techniques are difficult to apply. We tackle this by designing transformations that map a generic instance $(p, \Lambda, R)$ to a new one $(p', \Lambda', R')$ with a *larger Price of Fairness*. These include linearizing $R$ over subintervals of $[0, 1]$, averaging probabilities in $\Lambda$ via Karamata's inequality, and modifying $\Lambda$ so that the maximum-probability coordinate $c^* \in [C]$ of $p$ lies outside it, among others. Iteratively applying these transformations reduces the original problem, for fixed size $\lambda = |\Lambda|$, to a *single-variable* optimization problem solvable via first-order conditions, yielding $\psi_\lambda$, the worst-case value for that size. The final result is obtained by maximizing over all $\lambda$. See full proof in Appendix E.10. $\square$

We have reduced a complex optimization problem to a simple closed-form formula with a nice interpretation, the $\psi_\lambda$ corresponding to the worst-case price of opportunity fairness for a fixed size $\lambda = |\Lambda|$. It may intuitively seem that the Price of Fairness should always be 1, as randomly coloring the ground set is equivalent to first drawing a random number of agents per group and then placing them uniformly on the ground set, seemingly ensuring group-independent opportunities. Nonetheless, a striking conclusion from Theorem 4.8 is that **the price of opportunity fairness can still exceed 1, even when all agents are treated identically**. In particular, whenever $\max_{c \in [C]} p_c$ is larger than $1/2$, the worst-case PoF is at least $C - 2$, as observed in Figure 4.

On the other hand, Theorem 4.8 provides meaningful PoF bounds whenever $\max_{c \in [C]} p_c \leq 1/2$. More importantly, Theorem 4.8 immediately implies the following corollary.

**Corollary 4.9.** *Whenever* $\max_{c \in [C]} p_c \leq 1/(C - 1)$, $\mathrm{PoF}(M_n)$ *converges in probability towards 1.*

For matroid allocation problems in large markets, there is no loss of social welfare due to opportunity fairness as long as no group is overrepresented. This is quite striking as this may contradict the intuition that unfairness stems from the presence of small protected groups that must be catered to, sacrificing the welfare of larger groups. The above corollary shows that even with the presence of an arbitrarily small group, there might be no social welfare loss when being fair. Instead, it is the presence of a single overwhelming group which makes resources hard to fairly allocate, for the specific notion of opportunity fairness.

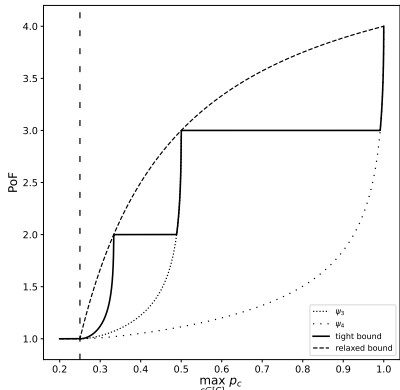

Figure 4: Theorem 4.8 for $C = 5$ groups: worst-case PoF, $\psi_\lambda$ for $\lambda \in \{3, 4\}$, and the relaxed bound $C - 1/\pi$.

We have shown how to bound the price of opportunity fairness in the semi-random setting by reducing the combinatorial optimization problem to make it tractable. However, considering the underlying matroid $(E, \mathcal{I})$ to be fixed may still be a pessimistic assumption for some real-world applications. For this reason, we study next a setting where both the matroid and the colors are drawn randomly.

## 4.4 Random Graphs Price of Fairness

A first approach to generating random matroids is to uniformly sample one among the $2^{2^{n-O(\log n)}}$ possible matroids on a ground set of size $n$, but this would mix different resource allocation problems. Instead, we focus on well-studied subclasses of matroids with established random models: graph-based matroids derived from Erdös-Rényi random graphs.

We consider *graphic matroids* on general graphs, where the ground set consists of edges, the independent sets are acyclic subgraphs, and a maximal allocation corresponds to the largest forest; and *transversal matroids* on bipartite graphs, where the ground set consists of nodes, the independent sets are bipartite matchings, and a maximal allocation corresponds to the largest matching —see Appendix B for the formal matroid class definitions. We show that whenever the edge probability between any two nodes is constant (i.e., independent of $n$), the price of opportunity fairness converges to 1 with high probability in both cases. Furthermore, by leveraging literature results on the structure of random graphs [37], these findings can be extended to settings with non-constant edge probability.

**Graphic matroid:** Given $n \in \mathbb{N}$ and $q \in [0,1]$, we consider the Erdös-Rényi random graph $G_{n,q} := ([n], A)$ with $n$ nodes such that for any $i, j \in [n]$, the edge $(i, j)$ belongs to $A$ independently with probability $q$. Given a random graph $G_{n,q} = ([n], A)$ and $p \in \Delta^C$, we consider a $p$ **randomly colored random graphic matroid**, denoted $G_{n,q}(p)$. Remark that the random coloring process and the random edges connections are done independently from one another.

**Proposition 4.10.** *Let $\omega = \omega(n)$ be a function such that $\omega(n) \to \infty$. Whenever $q \leq 1/(\omega n)$ or $q \geq \omega/n$, for any $p \in \Delta^C$, $\mathrm{PoF}(G_{n,q}(p))$ converges to 1 with high probability as $n$ grows.*

Thus, except near the critical threshold $q = \Theta(1/n)$ marking the transition from a forest to a giant component, almost all sufficiently large random graphic matroids incur no utility loss under opportunity fairness.

**Transversal matroid (matching):** Given $n \in \mathbb{N}$, $\beta \in (0,1)$ such that $\beta n \in \mathbb{N}$, and $q \in [0,1]$, we consider the random bipartite graph $B_{n,\beta,q} := ([n], [\beta n], A)$, where for any $i \in [n], j \in [\beta n]$, the edge $(i, j) \in A$ independently with probability $q$. Given a random Erdös-Rényi bipartite graph $B_{n,\beta,q}$ and $p \in \Delta^C$, we consider a $p$ **randomly colored random transversal matroid** denoted $B_{n,\beta,q}(p)$. Recall, the coloring process and the edges are drawn independently between them.

**Proposition 4.11.** *Let $\omega = \omega(n)$ be a function such that $\omega(n) \to \infty$ arbitrarily slow as $n \to \infty$. Whenever $q \leq 1/(\omega n^{3/2})$ or $q \geq \omega \log(n)/n$, for any $p \in \Delta^C$, $\mathrm{PoF}(B_{n,\beta,q}(p))$ converges to 1 with high probability as $n$ grows.*

As with graphic matroids, imposing opportunity fairness on random bipartite matching problems yields no utility loss when the edge probability is sufficiently small or large. Unlike the graphic case, however, the behavior in the intermediate regime near the phase transition remains unclear.

## 5 Conclusions and Limitations

We address matroid allocation problems under a novel group-fairness notion—opportunity fairness—, and prove tight bounds for the Price of Fairness, i.e., the loss of social welfare due to the fairness restrictions, in multiple settings from adversarial to fully random. Our model has two main limitations. First, integral allocations are only well approximated by fractional ones in large markets, which corresponds well to our motivating examples (e.g., job market) but may not always hold. Second, we considered the allocation cardinality as our notion of social welfare, i.e., each allocation has the same weight. The extension to weighted matching is straightforward if weights are assigned at the level of groups as it simply skews the polymatroid. However, individual-level weights would break the anonymity property and the polymatroid nature of $M$, hence new techniques would be required.

## Acknowledgments

This work has been supported by the French National Research Agency (ANR) through grant ANR-20-CE23-0007, ANR-23-CE23-0002, and PEPR IA FOUNDRY project (ANR-23-PEIA-0003). This work has also been funded by the European Union through the ERC OCEAN 101071601 grant. Views and opinions expressed are however those of the author(s) only and do not necessarily reflect those of the European Union or the European Research Council Executive Agency. Neither the European Union nor the granting authority can be held responsible.

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

# A  Further Related Works

**Fairness notions:** The study of fair algorithmic decision-making cover a broad range of fields, with fair division [64] in economics that focuses on concepts such as envy-freeness [16, 51, 66, 67] and maximin fairness [62], and machine learning that emphasizes statistical fairness notions like group fairness [23, 36, 60], including demographic parity [52]. Our new notion of opportunity fairness is inspired from both Equality of Opportunity [42] and Kalai-Smorodinsky fairness [47, 57]. Equality of Opportunity aims for *true-positive* rates to be independent of sensitive attributes; for opportunity fairness, this translates to ensuring that the resources allocated to a group are proportional to its opportunity level, the maximum allocation it could receive if it were considered in isolation. On the other hand, Kalai-Smorodinsky fairness requires maximizing the ratio $|S \cap E_c| / \max_{S' \in \mathcal{I}} |S' \cap E_c|$. While any maximum-size opportunity fair allocation satisfies Kalai-Smorodinsky fairness, the reverse does not necessarily hold, making our fairness notion more restrictive.

**The Price of Fairness:** The Price of Fairness was concurrently introduced by [11], who focus on maximin fairness and proportional fairness, and [15] who prioritize equitability and envy-freeness. They provided bounds for each fairness notion depending on the number of agents. Subsequently, [57] studied the price of fairness under the Kalai-Smorondinsky fairness notion for the *subset sum problem* and [29] studied the price of fairness in kidney-exchange. The concept of price of fairness has been extended to others research domains such as supervised machine learning [41, 39, 55] where the cost of fairness is studied on different prediction tasks. In this article, we initiate the study of the Price of Fairness under opportunity fairness. Unlike equitability, which requires identical allocations across groups, opportunity fairness is a more robust notion in the context of price of fairness. When some groups are inherently unable to receive resources, enforcing equitability leads to significant welfare loss, whereas we show that the price of fairness under opportunity fairness remains bounded. More importantly, we go beyond the traditional adversarial worst-case analysis, and instead consider more structured inputs, in the vein of [59], allowing for an average case analysis that better reflect trade-offs in real-world instances.

**Fair Matroid Allocation Problems:** The main objective of the matroid and fairness literature, initiated by [18], is to efficiently approximate maximum size fair allocations. Subsequent works extend this framework to submodular function optimization under fairness and matroid constraints [31, 32, 65, 68], while [9] study the computational complexity of finding optimal proportionally fair matching for more than two groups. We remark that maximum size opportunity fair fractional allocations can be computed efficiently whenever the underlying matroid possesses a polynomial-time independence oracle ([61], Chapter 44), as is the case of bipartite matching and communication network problems.

**Fair matching:** Recent work in matching have increasingly examined the impact of different fairness notions on matching mechanisms and how to design fair algorithms. [17, 28] and [48] examine the relation between fairness and stability in matching. Additionally, fairness in online matching has been studied in various contexts, including waiting time, equality of opportunity, and fairness constraints on the offline side of the market [19, 33, 45, 53, 60]. Our work contributes to this growing literature by introducing a new fairness notion, opportunity fairness, that captures the structural constraints of allocation problems.

# B  Matroid classes

The three examples introduced in Section 1 can be modeled as matroid allocation problems. Indeed,

1. The bipartite matching problem corresponds to a **transversal matroid**: Let $G = (U, V, A)$ be a bipartite graph. For a matching $\mu \subseteq A$ we denote $\mu(U) := \{u \in U \mid \exists v \in V, (u, v) \in \mu\}$. The pair $(U, \mathcal{I})$, with $\mathcal{I} := \{\mu(U), \forall \mu \subseteq A \text{ matching}\}$, is called a transversal matroid.

2. The research funding allocation problem corresponds to a **partition matroid**. Let $E$ be a finite set partitioned into disjoint subsets $E^1, E^2, \ldots, E^k$, each representing, for instance, a department or category of funding. Given nonnegative integers $b_1, b_2, \ldots, b_k$, the partition matroid is the pair $(E, \mathcal{I})$, where $\mathcal{I} := \{S \subseteq E \mid |S \cap E^i| \leq b_i \text{ for all } i = 1, \ldots, k\}$. This formulation captures allocation constraints where each group (e.g., department) has its own capacity limit.

3. The faculty assignment problem corresponds to a **laminar matroid**. Let $E$ be a finite set and $\mathcal{L}$ a laminar family of subsets of $E$ (that is, for any $L_1, L_2 \in \mathcal{L}$, either $L_1 \subseteq L_2$, $L_2 \subseteq L_1$, or $L_1 \cap L_2 = \emptyset$). Given a capacity function $b : \mathcal{L} \to \mathbb{N}$, the laminar matroid is the pair $(E, \mathcal{I})$, where

$\mathcal{I} := \{S \subseteq E \mid |S \cap L| \leq b(L) \text{ for all } L \in \mathcal{L}\}$. This structure models hierarchical constraints, such as departmental and team-level capacity limits in faculty assignments, or in private companies.

We also give the definition of further matroid classes, which are mathematically relevant to the studied problem.

4. **Graphic matroid**: Let $G = (U, A)$ be a graph. The pair $(A, \mathcal{I})$, with $\mathcal{I} := \{S \subseteq A \mid S \text{ is acyclic}\}$, is called a graphic matroid.

5. **Uniform matroid**: Let $E$ be a finite set and $b \in \mathbb{N}$. The $b$-uniform matroid is the pair $(E, \mathcal{I})$, such that, $\mathcal{I} := \{S \subseteq E \mid |S| \leq b\}$.

## C   Null integral opportunity fair allocations

Figure 5 illustrates an example on a graphic matroid[4] for two groups. The figure on the right shows the integer feasible allocations (the blue dots) and the set of opportunity fair allocations (the orange line), whose only intersection is at the origin.

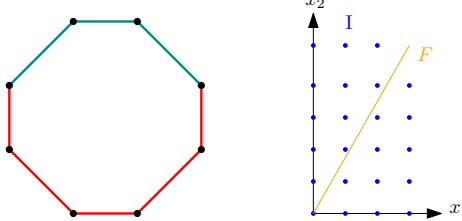

Figure 5: Graphic matroid example showing that integrality can lead to null opportunity fair allocations. (Left) Graph defining a colored graphic matroid with two groups. Group 1 is denoted by the green edges while Group 2 by the red edges. It follows that $r(1) = 5$, $r(2) = 3$, and $r(\{1, 2\}) = 7$. (Right) Set of integer feasible allocations (blue dots) and set of opportunity fair allocations (orange line), whose only intersection is at the origin.

## D   Efficient computation of Opportunity fair allocations of maximum size

Whenever a matroid $(E, \mathcal{I})$ possesses a poly-time independent oracle, that is, an oracle that for any subset $S$ of $E$ tells if $S$ is an independent set or not in polynomial time, we can construct a poly-time separation oracle for the corresponding polymatroid in $\mathbb{R}^{|E|}$ [61][Theorem 40.4]. A separation oracle, given a vector $x \in \mathbb{R}^{|E|}$ and the polymatroid

$$Q := \left\{ x \in \mathbb{R}_+^{|E|} \mid \sum_{e \in S} x_e \leq \text{rank}(S), \forall S \subseteq E \right\},$$

either tells if $x$ belongs to $Q$ or outputs a separating hyperplane. Starting from the poly-time separation oracle of a matroid, we can obtain a poly-time separation oracle for set of feasible fractional allocation of the corresponding colored matroid (or polymatroid). Therefore, we can compute an opportunity fair allocation of maximum size by solving the following problem,

$$\max \sum_{c \in [C]} x_c$$

$$\text{s.t. } x_c = \sum_{e \in E : e \in c} y_e \qquad \forall c \in [C]$$

$$\frac{x_c}{\text{rank}(c)} = \frac{x_{c'}}{\text{rank}(c')}, \qquad \forall c, c' \in [C]$$

$(y_e)_{e \in E}$ is a feasible fractional allocation

$$x \in [0, 1]^C, y \in [0, 1]^{|E|}.$$

Indeed, the previous problem can be solved in polynomial time using the ellipsoid method provided with the poly-time separation oracle of the polymatroid and the (trivial) poly-time separation oracle of the polytope of fairness constraints.

---

[4]Allocations within a graphic matroid correspond to acyclic subgraphs of the given graph.

One possible downside of this approach, is that the ellipsoid method can be quite slow in practice. Nonetheless, for many common matroid classes, including transversal and graphic matroids, the fair optimization problem admits a polynomial-size linear programming formulation, which allows us to use more efficient algorithms such as Simplex or interior point methods.

Indeed, this relates to the concept of extension complexity, which basically refers to the minimal number of constraints needed to describe the independence polytope, possibly adding some variables. Notably, [7] show that regular matroids have polynomial size extension complexity.

# E   Missing proofs

## E.1   Proof of Proposition 2.4

**Proposition 2.4.** For any fractional opportunity fair maximum size allocation $x \in F$, there always exists a random allocation $X$, such that, $\mathbb{E}[X] = x$ and for all realizations, $X$ is feasible, integral, $\left(1 - \frac{2C}{\min_c r(c)}\right)$-opportunity fair, and $\|X - x\|_1 \leq C$.

*Proof.* Let $M$ be a set of fractional allocations and $x$ be an optimal fair allocation. Consider the translated unit cube $K = \lfloor x \rfloor + [0, 1]^C$ where $\lfloor x \rfloor = (\lfloor x_1 \rfloor, \ldots, \lfloor x_C \rfloor)$, which contains $x$ and all the integer vertices closest to $x$. We aim to prove that $K \cap M$ yields integer extreme points.

Let $M' := \mathbb{R}_+^C \cap (M - \lfloor x \rfloor)$ to be the positive quadrant of $M$ translated by $\lfloor x \rfloor$. It holds

$$M' = \big\{ y \in \mathbb{R}_+^C \mid \sum_{c \in \Lambda} y_c + \lfloor x_c \rfloor \leq r(\Lambda), \forall \Lambda \subset [C] \big\}$$

$$= \big\{ y \in \mathbb{R}_+^C \mid \sum_{c \in \Lambda} y_c \leq r(\Lambda) - \sum_{c \in \Lambda} \lfloor x_c \rfloor, \forall \Lambda \subset [C] \big\}.$$

Consider the function $f(\Lambda) := r(\Lambda) - \sum_{c \in \Lambda} \lfloor x_c \rfloor$. Since $x \in M$, so is $\lfloor x \rfloor$, hence $f(\Lambda) \geq 0$, and $f$ takes only integer values as both $r$ and $\lfloor x \rfloor$ take integer values. Moreover, $f$ is submodular as both $r$ and the application $\Lambda \mapsto \sum_{c \in \Lambda} x_c$ are submodular. Therefore, $M'$ is an integral polymatroid.

The unit cube being also an integral polymatroid, by Theorem 35 of [30], $[0, 1]^C \cap M'$ has integral vertices, and thus so it does,

$$\lfloor x \rfloor + ([0, 1]^C \cap M') = K \cap M.$$

It follows that, since $x \in K \cap M$, which is a convex set (as both $K$ and $M$ are convex sets), $x$ can be described as the convex combination of the extreme points of $K \cap M$. Since all extreme points of $K \cap M$ are integral, they are, in particular, a subset of the extreme points of $K$. By Carathéodory's Theorem, there exists $X^{(1)}, \ldots, X^{(k)}$ vertices of $K \cap M$ for $k \leq C + 1$, and $(p_1, \ldots, p_k) \in [0, 1]^{C+1}$ with $\sum_{i=1}^k p_i = 1$, such that $x = \sum_{i=1}^k p_i X^{(i)}$.

The randomized rounding $X$ then is defined by drawing $X^{(i)}$ with probability $p_i$. Note that for any realization of $X^{(i)}$ of $X$, $X^{(i)}$ is feasible and integral by construction. Moreover,

$$\mathbb{E}[X] = \sum_{i=1}^k p_i X^{(i)} = x,$$

and since $X^{(i)}$ for any $i \in \{1, ..., k\}$, and $x$ belong to $[0, 1]^C$, it always follows that $\|X^{(i)} - x\|_1 \leq C$. Regarding the $\gamma$-opportunity fairness, recall that $x$ is 1-opportunity fair, so there exists $\alpha \in [0, 1]$ such that $x = \alpha(r(1), \ldots, r(C))$. Moreover, because $x$ is optimal, $\alpha \geq 1/C$. Indeed, either PoF $= 1 = r([C])/(\alpha \sum_c r(c))$ which yields $\alpha = r([C])/\sum_c r(c) \geq r([C])/(Cr([C])) = 1/C$, or PoF $> 1$ in which case $(r(1), \ldots, r(C))/(C - 1)$ is feasible (see proof of Theorem 4.1) which implies, by definition, that $\alpha \geq (C - 1)$. For any $i, j \in [C]$, it follows,

$$\frac{X_i/r(i)}{X_j/r(j)} \geq \frac{(x_i - 1)/r(i)}{(x_j + 1)/r(j)} \geq \frac{\alpha - 1/\min_c r(c)}{\alpha + 1/\min_c r(c)} = 1 - \frac{2}{\alpha \min_c r(c) + 1} \geq 1 - \frac{2C}{\min_c r(c)},$$

concluding the first part of the proof.

**Worst-case $L_1$ deviation:** Consider a submodular function $f$ such that for $S \subset [C]$, $f(S) = \max(|S|, C \cdot \mathbf{1}[1 \in S])$. This submodular function induces a integral polymatroid $M$ (hence a colored matroid by surjection). The Pareto front of this polymatroid is $P = \{x \in \mathbb{R}_+^C : x_i \in [0, 1] \text{ for } 2 \leq i \leq C \text{ and } x_1 = C - \sum_{i \geq 2} x_i\}$. The optimal fair allocation is $x = (C^2/(2C - 1), C/(2C - 1), ..., C/(2C - 1))$. We now show that $x$ is far from all vertices. Let $y$ be an integral point of $M$, and let $p$ be the projection over the last $C - 1$ coordinates. Because $p$ is a projection, $\|x - y\|_1 \geq \|p(x) - p(y)\|_1$. Now $p(x) = C/(2C - 1) \cdot \mathbf{1}_{C-1}$ lies in the unit cube $[0, 1]^{C-1}$, and the closest vertex is $y' = \mathbf{1}_{C-1}$. Overall,

$$\|x - y\|_1 \geq \|p(x) - p(y)\|_1 \geq \|C/(2C-1) \cdot \mathbf{1}_{C-1} - \mathbf{1}_{C-1}\|_1 = (C-1)\frac{C-1}{2C-1} \geq \frac{C}{2} - \frac{1}{4} = \Omega(C),$$

as $C \geq 2$. This is true for any vertex $y$, hence no rounding scheme $X$ can do better.

**Worst-case fairness degradation:** Using the construction from fig. 6, suppose group 1 matches up to $r_1$ agents, and the remaining groups share $r_2$ agents, with $(r_1/(C-1), r_2/(C-2), \ldots, r_2/(C-2))$ the optimal fair allocation. Let $r_1$ and $r_2$ be such that the nearest integers to $r_1/(C-1)$ and $r_2/(C-2)$ are respectively $r_1/(C-1) - 1/2 + \epsilon$ and $r_2/(C-2) + 1/2 - \epsilon$. Hence, for $(C-1) \leq r_2 << r_1$, we have

$$\frac{(r_1/(C-1) - 1/2)/r_1}{(r_2/(C-1) + 1/2)/r_2} = \frac{1/(C-1) - 1/(2r_1)}{1/(C-1) + 1/(2r_2)} = 1 - \frac{1/(2r_1) + 1/(2r_2)}{1/(C-1) + 1/(2r_2)}$$

$$\approx 1 - \frac{C-1}{C - 1 + 2r_2} \leq 1 - \frac{C-1}{3r_2} = 1 - O\left(\frac{C}{\min_c r(c)}\right).$$

So no integral allocation can have a better approximately fair guarantee than $1 - O(C/\min_c r(c))$.

$\square$

### E.2 Proof of Proposition 2.5

**Proposition 2.5.** Let $M$ be a $C$-colored matroid. Then, either $\gamma < \max_{x \in \mathrm{F}} \min_{c \in [C]} x_c/r(c)$ and $\mathrm{PoF}_\gamma(M) = 1$, or

$$\gamma \cdot \mathrm{PoF}(M) \leq \mathrm{PoF}_\gamma(M) \leq \frac{\gamma \cdot \mathrm{PoF}(M)}{1 - (1 - \gamma)^{\frac{((C-1) - \mathrm{PoF}(M))}{C-2}}}.$$

*Proof.* The lower bound arises by disregarding the feasibility constraints, and improving the original optimal fair solution by simply scaling non-tight coordinates by $\gamma$. The upper bound is obtained via a convex combination between the optimal fair and optimal unfair solution which is made to be exactly $\gamma$-fair, and which remains feasible by convexity.

**Lower bound.** Let $x = \mathrm{argmax}_{y \in \mathrm{F}} \|y\|_1 = \alpha(r(1), ..., r(C))$, where $\alpha \in [0, 1]$. If $x$ is maximal, $\mathrm{PoF}_\gamma(M) = \mathrm{PoF}(M) = 1$. Otherwise, there exists $\Lambda \subsetneq [C]$ such that $x \in \mathrm{argmax}_{y \in M} \sum_{c \in \Lambda} y_c$. Define $x'$ as $x_i' = x_i$ for all $i \in \Lambda$ and $x_i' = \frac{1}{\gamma}x_i$ for all $i \notin \Lambda$.

Let us denote by $F_\gamma$ the $\gamma$-opportunity fair feasible set. First, we prove that $\|x'\|_1 \geq \max_{F_\gamma} \|x\|_1$. Suppose there exists $x'' \in F_\gamma$ a $\gamma$-opportunity realizable point such that $\|x''\|_1 > \|x'\|_1$. Let $\Gamma = \{i \in [C], x_i'' > x_i'\}$. Since

$$\sum_\Lambda x_i'' \leq r(\Lambda) = \sum_\Lambda x_i',$$

it follows that, $\sum_{\Lambda^c} x_i'' > \sum_{\Lambda^c} x_i'$, and thus $\Gamma \cap \Lambda^c \neq \emptyset$. Let $i \in \Gamma \cap \Lambda^c$, then $x_i'' > x_i' = \frac{1}{\gamma}x_i$. Moreover, there must exist $j \in \Lambda$ such that $x_j'' \leq x_j$ since $x \in \mathrm{argmax}_M \sum_{c \in \Lambda} y_c$. Therefore,

$$\frac{x_i''}{r(i)} > \frac{1}{\gamma} \cdot \frac{x_i}{r(i)} = \frac{1}{\gamma} \cdot \frac{x_j}{r(j)} \geq \frac{1}{\gamma} \cdot \frac{x_j''}{r(j)},$$

which contradicts $x''$ being $\gamma$-opportunity fair. It follows,

$$\|x'\|_1 = \sum_{i \in \Lambda} x_i + \frac{1}{\gamma} \sum_{i \in \Lambda^c} x_i = \alpha\left(\sum_{i \in \Lambda} r(i) + \frac{1}{\gamma} \sum_{i \in \Lambda^c} r(i)\right),$$

and therefore,

$$\frac{\text{PoF}(M)}{\text{PoF}_\gamma(M)} = \frac{\max_{y \in F_\gamma} \|y\|_1}{\max_{y \in F} \|y\|_1} \leq \frac{\|x'\|_1}{\|x\|_1} = \frac{\sum_{i \in \Lambda} r(i) + \frac{1}{\gamma} \sum_{i \in \Lambda^c} r(i)}{\sum_{i \in [C]} r(i)} \leq \frac{1}{\gamma},$$

thus, $\text{PoF}_\gamma(M) \geq \gamma \text{PoF}(M)$.

**Upper bound**. As before, let $x = \text{argmax}_{y \in F} \|y\|_1 = \alpha(r(1), ..., r(C))$. We suppose that $x$ is not optimal, otherwise $\text{PoF}_\gamma = \text{PoF} = 1$. Let $x'$ be a point in the Pareto frontier that Pareto dominates $x$. By the polymatroid characterization, it holds that $x$ is minimal, i.e. $\|x\|_1 = r([C])$. For $\lambda \in \mathbb{R}_+$, consider the combination $z = \lambda x' + (1 - \lambda)x$. Because $x'$ Pareto dominates $x$, we have $x'_i \geq x_i$ for all $i \in [C]$, hence $z_i \geq x_i$. By definition of $r(i)$ we also have that $x'_i \leq r(i)$, thus $z_i \leq \lambda r(i) + (1 - \lambda)x_i$. This implies that

$$\frac{z_i/r(i)}{z_j/r(j)} \geq \frac{x_i/r(i)}{\lambda + (1 - \lambda)(x_j/r(j))} = \frac{\alpha}{\lambda + (1 - \lambda)\alpha}.$$

In particular, $z$ is $\gamma$-opportunity fair whenever $\alpha/(\lambda + (1 - \lambda)\alpha) \geq \gamma$, i.e., whenever $\lambda \leq \frac{\alpha(1-\gamma)}{\gamma(1-\alpha)}$. If $\alpha(1 - \gamma)/(\gamma(1 - \alpha)) > 1$, which is equivalent to $\gamma < \alpha$, then taking $\lambda = 1$ yields that $x'$ is $\gamma$-fair, and it is already optimal so $\text{PoF}_\gamma = 1$. In the following we let $\gamma \geq \alpha$. Let us take $\lambda = \alpha(1 - \gamma)/(\gamma(1 - \alpha)) \leq 1$ such that $z$ in $M$ by convex combination. Because $z$ is $\gamma$-opportunity fair and feasible, $z \in F_\gamma$. Hence

$$\text{PoF}_\gamma(M) = \frac{r([C])}{\max_{y \in F_\gamma} \|y\|_1} \leq \frac{r([C])}{\|z\|_1} = \frac{r([C])}{\lambda \sum_{c \in [C]} x'_c + (1 - \lambda) \sum_{c \in [C]} x_c}$$

$$= \frac{\text{PoF}(M)}{\lambda \text{PoF}(M) + (1 - \lambda) \cdot 1}$$

$$= \frac{\text{PoF}(M)}{\frac{\alpha(1-\gamma)}{\gamma(1-\alpha)}\text{PoF}(M) + \frac{\gamma(1-\alpha) - \alpha(1-\gamma)}{\gamma(1-\alpha)}}$$

$$= \frac{\gamma(1 - \alpha)\text{PoF}(M)}{\alpha(1 - \gamma)\text{PoF}(M) + \gamma - \alpha}$$

$$\leq \frac{\gamma(1 - \frac{1}{C-1})\text{PoF}(M)}{\frac{1}{C-1}(1 - \gamma)\text{PoF}(M) + \gamma - \frac{1}{C-1}}$$

$$= \frac{\gamma\text{PoF}(M)}{1 - (1 - \gamma)\frac{((C-1)-\text{PoF}(M))}{C-2}},$$

where we used that $\alpha \geq 1/(C - 1)$ as otherwise $\text{PoF}(M) = 1$. $\qquad\square$

### E.3 Proof of Proposition 3.2

**Proposition 3.2.** Let $((E_c)_{c \in [C]}, \mathcal{I})$ be a colored matroid with rank function $r$ and set of feasible fractional allocations $M$. Then, $M$ is the polymatroid associated to the function $r$, i.e.,

$$M = \Big\{ x \in \mathbb{R}_+^C \mid \sum_{c \in \Lambda} x_c \leq r(\Lambda), \forall \Lambda \subseteq [C] \Big\}.$$

*Proof.* Let $((E_c)_{c \in [C]}, \mathcal{I})$ be a $C$-colored matroid. It is sufficient to show that

$$I := \Big\{ (|S \cap E_1|, \ldots, |S \cap E_C|) \in \mathbb{N}^C \mid S \in \mathcal{I} \Big\},$$

the set of integer feasible allocation, is a discrete polymatroid, and to conclude by taking the convex hull, as an integral polymatroid is equivalent to the convex hull of a discrete polymatroid [30]. We prove this by showing that I satisfies equivalent conditions for a set to be a discrete polymatroid [44], which are:

1. For any $x \in \mathbb{N}^C$ and $y \in I$ such that $x \leq y$ (component wise), $x \in I$.
2. For any $x, y \in I$, with $\|x\|_1 < \|y\|_1$, there exists $c \in [C]$ such that $x_c < y_c$ and $x + \vec{e}_c \in I$, where $\vec{e}_c$ is the canonical vector with value 1 at the $c$-th entry and 0 otherwise.

We will repeatedly leverage the properties of the underlying matroid to show, by exhaustion, that it is always possible to find an element in the allocation associated with $y$ whose color is underrepresented in the allocation associated with $x$.

The first property is a direct consequence of the hereditary property of $\mathcal{I}$. Let $x, y \in \mathrm{I}$ such that $\|x\|_1 < \|y\|_1$. Let $A_x, A_y \in \mathcal{I}$ be two independent sets such that $(|A_x \cap E_1|, \ldots, |A_x \cap E_C|) = x$ and $(|A_y \cap E_1|, \ldots, |A_y \cap E_C|) = y$. In particular, $|A_x| < |A_y|$. By the augmentation property, there exists $e \in A_y \setminus A_x$ such that $A_x \cup \{e\} \in \mathcal{I}$. Let $c$ be the group of $e$. If $x_c < y_c$, the proof is over. Suppose, otherwise, that $x_c \geq y_c$. Let $e' \in A_x$ such that $e' \in E_c$ as well, and define $A'_x = A_x \cup \{e\} \setminus \{e'\}$. By the hereditary property, $A'_x \in \mathcal{I}$ and, by construction, $(|A'_x \cap E_1|, \ldots, |A'_x \cap E_C|) = x$. Applying again the augmentation property, there exists $e'' \in A_y \setminus A'_x$ such that $A'_x \cup \{e''\} \in \mathcal{I}$. Notice that $e'' \notin \{e, e'\}$. The proof is concluded by repeating the same argument until finding an element in some $E_c$ such that $x_c < y_c$. The procedure stops after a finite amount of iteration as at every iteration the element in $A_y$ obtained from the augmentation property must be different to all previous ones as well to all replaced elements in $A_x$. $\qquad\square$

### E.4  Proof of Corollary 3.3

**Corollary 3.3.** The price of opportunity fairness of a polymatroid $M$ is given by,

$$\mathrm{PoF}(M) = \frac{\mathrm{r}([C])}{\sum_{c \in [C]} \mathrm{r}(c)} \cdot \max_{\Lambda \subseteq [C]} \frac{\sum_{c \in \Lambda} \mathrm{r}(c)}{\mathrm{r}(\Lambda)}.$$

*Proof.* Let $x^*$ be a maximum size opportunity fair allocation. The opportunity fair requirement implies that $x^*$ belongs to the line $t \cdot (\mathrm{r}(1), \ldots, \mathrm{r}(C))$ for $t > 0$. Let $t^* > 0$ such that $x^* = t^* \cdot (\mathrm{r}(1), \ldots, \mathrm{r}(C))$. Since $x^*$ is a feasible fractional allocation, Proposition 3.2 implies that for any $\Lambda \subseteq [C]$,

$$t^* \sum_{c \in \Lambda} \mathrm{r}(c) \leq \mathrm{r}(\Lambda).$$

It follows that

$$t^* = \min_{\Lambda \subseteq [C]} \frac{\mathrm{r}(\Lambda)}{\sum_{c \in \Lambda} \mathrm{r}(c)},$$

and, in particular, that,

$$\mathrm{PoF}(M) = \frac{\mathrm{r}([C])}{t^* \sum_{c \in [C]} r(c)} = \frac{\mathrm{r}([C])}{\sum_{c \in [C]} \mathrm{r}(c)} \cdot \max_{\Lambda \subseteq [C]} \frac{\sum_{c \in \Lambda} \mathrm{r}(c)}{\mathrm{r}(\Lambda)},$$

concluding the proof. $\qquad\square$

### E.5  Proof of Theorem 4.1

**Theorem 4.1.** For any $C$-colored matroid $M$, we have $\mathrm{PoF}(M) \leq C - 1$, and this bound is tight.

*Proof.* Using the monotonicity and the sub-additivity of the colored-matroid rank function $\mathrm{r}$, we will upper bound the alternate formulation of the PoF in Equation (1).

Let $M$ be a $C$-dimensional polymatroid. Let $\Lambda^*$ be the maximizer of Equation (1). Whenever $\Lambda^* = [C]$ it holds $PoF(M) = 1 \leq C - 1$. Suppose then that $\Lambda^* \subsetneq [C]$. It follows,

$$\mathrm{PoF}(M) = \frac{\mathrm{r}([C])}{\mathrm{r}(\Lambda^*)} \cdot \frac{\sum_{c \in \Lambda^*} \mathrm{r}(c)}{\sum_{c \in [C]} \mathrm{r}(c)}.$$

Since $\mathrm{r}(\Lambda^*) \geq \max_{c \in \Lambda^*} \mathrm{r}(c) \geq \frac{1}{|\Lambda^*|} \sum_{c \in \Lambda^*} \mathrm{r}(c)$, it follows that,

$$\mathrm{PoF}(M) \leq |\Lambda^*| \cdot \frac{\mathrm{r}([C])}{\sum_{c \in \Lambda^*} \mathrm{r}(c)} \cdot \frac{\sum_{c \in \Lambda^*} \mathrm{r}(c)}{\sum_{c \in [C]} \mathrm{r}(c)} \leq |\Lambda^*| \leq C - 1,$$

where we have used that $\mathrm{r}([C]) \leq \sum_{c \in [C]} \mathrm{r}(c)$ because $r$ is submodular and thus sub-additive, and $|\Lambda^*| \leq C - 1$ as $\Lambda^* \neq [C]$.

To show that this bound is tight, we exhibit a sequence of $C$-dimensional polymatroids for which the bound is tight in the limit. Consider a bipartite graph as in Figure 6. Let $E_1$ be independently

connected to $r_1$ nodes, and $E_2, \ldots, E_C$ be completely connected to the same $r_2$ nodes. The last $C - 1$ groups are in competition for resources while the first group suffers no competition. It holds $\text{r}(1) = r_1$, while for any $\Lambda \subseteq [C] \setminus \{1\}$, $\text{r}(\Lambda) = r_2$, and $\text{r}(\Lambda \cup \{1\}) = r_1 + r_2$. In particular, it follows that $\Lambda^*$, the maximizer of Equation (1), is given by $\Lambda^* = [C] \setminus \{1\}$, and,

$$\text{PoF}(M) = \frac{r_1 + r_2}{r_1 + (C-1)r_2} \cdot \frac{(C-1)r_2}{r_2} = \frac{r_1 + r_2}{r_1 + (C-1)r_2} \cdot (C-1) \xrightarrow[r_1 \to \infty]{} C - 1. \quad \square$$

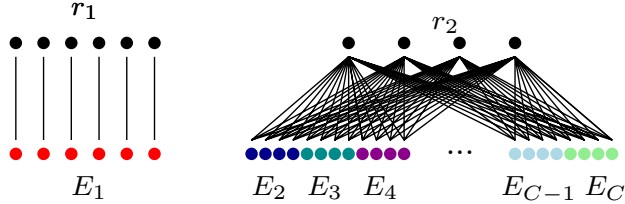

Figure 6: Transversal polymatroid that makes the PoF bound of $C - 1$ tight (Theorem 4.1). Group 1 is totally independent of the rest of the groups, while groups $\{2, 3, ..., C\}$ compete for the same resources.

The proof of Theorem 4.1 shows that for transversal matroid, i.e. bipartite matching, our main application, the bound is tight. Regarding **graphic matroids** (see Appendix B for a definition) , the Walecki construction [6] which states that any clique of $2C - 1$ vertices has a decomposition into $C - 1$ disjoint Hamiltonian cycles, allows to design a tight example for the PoF upper bound. Indeed, associate each Hamiltonian cycle to a color $c \in [C - 1]$ and add one extra group of edges with color $C$ as illustrated in Figure 7. It is not hard to see that this construction achieves a PoF equal to $C - 1$.

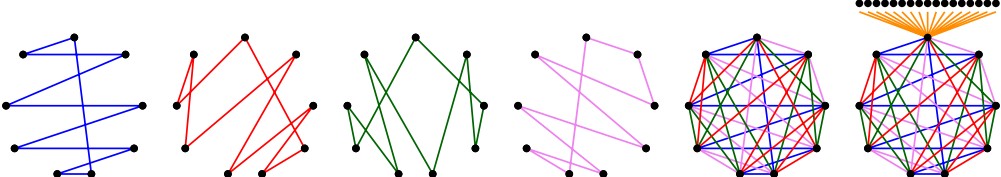

Figure 7: Graphic polymatroid tight example for worst-case PoF equal to $C - 1$ (Theorem 4.1), based on the Walecki construction, for 5 groups. Groups are represented by colored edges. The first 4 figures show the edges of each group, while the clique corresponds to their union. The final figure considers a 5-th group that is totally independent on the rest, leading to a similar construction as in Figure 6.

The same transversal matroid example can be used for **partition matroid**. This also implies the tightness of this bound for larger sub-classes of matroid which include either graphic or partition matroids, such as **linear** or **laminar matroids**. For **uniform matroids**, in exchange, it is immediate to see that $\text{PoF}(M) = 1$: if the opportunity fairness constraint is violated, it is always possible to take the excessive (potentially fractional) resources from over-represented colors and give them to the under-represented ones. It remains open to prove if intermediate cases (not 1 nor $C - 1$) exist for some family of matroids.

### E.6 Alternative proof of Corollary 4.2

Corollary 4.2 states that no loss of social welfare is incurred by imposing opportunity fairness when agents are divided into two groups. The same conclusion can be easily proven from a geometric point of view, as the line directed by $(r_1, r_2)$ necessarily intersects with the Pareto frontier, which corresponds to the set of social welfare maximizing allocations and therefore, it is directed by $(1, -1)$. For $C > 2$, the property does not hold anymore, as proven by the tight $C - 1$ bound, since the line directed by $(r(1), ..., r(C))$ does not necessarily intersect with the Pareto frontier. Figure 8 illustrates these situations for $C = 2$ and $C = 3$.

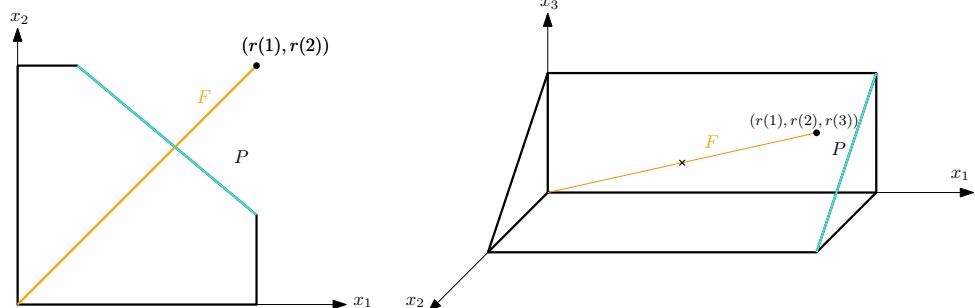

Figure 8: Relation between the Pareto frontier $P$ (light blue) and the set of opportunity fair allocations $F$ (orange) for two and three groups. For $C = 2$ they always intersect, however it is not always the case for $C > 2$ as illustrated on the right (the cross marks the largest feasible fair matching).

### E.7 Proof of Proposition 4.4

**Proposition 4.4.** For any $C$-colored matroid $M$, it holds,

$$\text{PoF}(M) \leq \frac{1}{2} \cdot \frac{\max_{c \in [C]} \mathrm{r}(c)}{\min_{c \in [C]} \mathrm{r}(c)} + \frac{C}{4} \cdot \left( \frac{\max_{c \in [C]} \mathrm{r}(c)}{\min_{c \in [C]} \mathrm{r}(c)} \right)^2 + \frac{1}{4C} \cdot \mathbb{1}\{C \text{ odd}\}.$$

Moreover, whenever all groups have the same rank, the resulting bound is tight.

*Proof.* From Corollary 3.3 we know that $\text{PoF}(M) = \frac{\mathrm{r}([C])}{\sum_{c \in [C]} \mathrm{r}(c)} \cdot \max_{\Lambda \subseteq [C]} \frac{\sum_{c \in \Lambda} \mathrm{r}(c)}{\mathrm{r}(\Lambda)}$. Let $\Lambda^*$ be the argmax in the equation, and let us reorder the groups so $\Lambda^* = [\ell]$ for some $\ell \in [C]$. Denote $\gamma := \frac{\max_{c \in [C]} \mathrm{r}(c)}{\min_{c \in [C]} \mathrm{r}(c)}$. First, remark that the sub-additivity of the r function (consequence of non-negativity and submodularity) implies the following inequality,

$$\mathrm{r}([C]) - \mathrm{r}([\ell]) \leq \mathrm{r}([C] \setminus [\ell]) \leq \sum_{c=\ell+1}^{C} \mathrm{r}(c) \leq (C - \ell) \max\{\mathrm{r}(c), c \in [C]\}.$$

It follows that

$$\text{PoF}(M) = \frac{\mathrm{r}([C])}{\mathrm{r}([\ell])} \cdot \frac{\sum_{c \in [\ell]} \mathrm{r}(c)}{\sum_{c \in [C]} \mathrm{r}(c)} \leq \left( 1 + \frac{\mathrm{r}([C]) - \mathrm{r}([\ell])}{\mathrm{r}([\ell])} \right) \frac{\ell}{C} \gamma \leq (1 + (C - \ell)\gamma) \frac{\ell}{C} \gamma.$$

The right-hand side of the previous inequality is maximized (subject to $\ell \in \mathbb{N}$) at $\ell = C/2 + \mathbb{1}\{C \text{ odd}\}/2\gamma$, which leads to the stated upper bound. Concerning the general result of the tightness of the bound, consider the constructions illustrated in Figures 9 and 10 where $\lfloor \frac{C}{2} \rfloor$ groups are isolated and $\lceil \frac{C}{2} \rceil$ compete for the same resources, respectively for transversal and graphic matroids, for (left) $C = 4$ and $\mathrm{r}(c) = 3$ for all $c \in [C]$, and (right) $C = 5$ with $\mathrm{r}(c) = 5$ for all $c \in [C]$. Suppose, moreover, that all groups have rank $r$, for some $r \in \mathbb{N}$. An opportunity fair allocation must allocate at most $r/\lceil \frac{C}{2} \rceil$ resources to each group. It follows that

$$\text{PoF} = \frac{Cr/\lceil \frac{C}{2} \rceil}{r + r \lfloor \frac{C}{2} \rfloor} = \frac{(1 + \lfloor C/2 \rfloor)\lceil C/2 \rceil}{C} = \frac{1}{2} + \frac{C}{4} + \frac{\mathbb{1}\{C \text{ odd}\}}{4C}. \qquad \square$$

### E.8 Proof of Proposition 4.6

**Proposition 4.6.** Let $M$ be a $C$-colored matroid. Suppose that for any $c, c'$ in $[C]$, $\mathrm{r}(c) = \mathrm{r}(c')$. Whenever $\rho \in [1/C, 1/(C-1)]$, $\text{PoF}(M) = 1$. Otherwise,

$$\text{PoF}(M) \leq \rho \max \left( \frac{C - \lfloor C\rho \rfloor + 1}{C\rho - \lfloor C\rho \rfloor + 1}, C - \lfloor C\rho \rfloor \right) \leq \rho((1 - \rho)C + 1).$$

In addition, the first upper bound is jointly tight in $\rho$ and $C$.

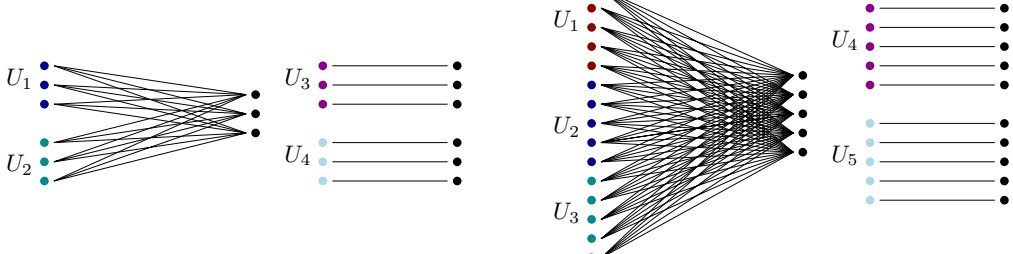

Figure 9: Tight bound example for PoF as stated in Proposition 4.4 for transversal matroids with (left) four groups and (right) five groups.

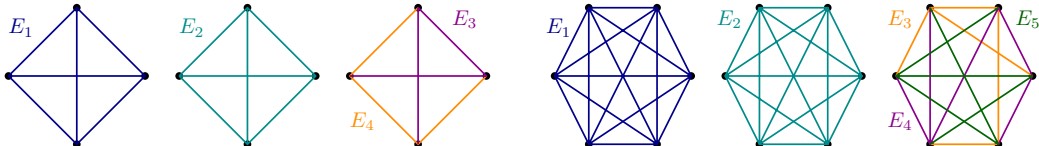

Figure 10: Tight bound example for PoF as stated in Proposition 4.4 for graphic matroids with (left) four groups and (right) five groups.

*Proof.* Let $M$ be a polymatroid such that all groups $c \in [C]$ have the same rank r. Suppose $\rho \in [1/C, 1/(C-1)]$ and $\mathrm{PoF}(M) > 1$. Bounding the PoF as in Theorem 4.1 plus using the fact that $\rho \leq 1/(C-1)$ leads to,

$$\mathrm{PoF}(M) \leq \rho(C-1) \leq 1,$$

which is a contradiction.

Suppose $\rho \in [1/(C-1), 1]$. Let $\alpha^* = \max\{\alpha \in [0, 1] : \alpha(\mathrm{r}, ..., \mathrm{r}) \in M\}$, the price of fairness is written as,

$$\mathrm{PoF} = \frac{\mathrm{r}([C])}{\alpha^* \sum_{c \in [C]} \mathrm{r}} = \frac{\rho \sum_{c \in [C]} \mathrm{r}}{\alpha^* \sum_{c \in [C]} \mathrm{r}} = \frac{\rho}{\alpha^*}.$$

Therefore, the stated upper bound comes from proving that

$$\frac{1}{\alpha^*} \leq \max\left\{\frac{C - \lfloor C\rho \rfloor + 1}{C\rho - \lfloor C\rho \rfloor + 1}, C - \lfloor C\rho \rfloor\right\}. \tag{5}$$

Let $\sigma \in \Sigma([C])$ be a permutation, $c \in [C]$, and denote, for $\ell \in [C]$,

$$\alpha_\ell(\sigma) := \frac{\mathrm{r}(\sigma([\ell]))}{\sum_{t \in [\ell]} \mathrm{r}(\sigma(t))},$$

where $\mathrm{r}(\sigma([\ell])) = \mathrm{r}(\{\sigma(1), ..., \sigma(\ell)\})$ corresponds to the size of a maximum size allocation in the submatroid obtained by restricting to the groups in the first $\ell$ entries of $\sigma$. With this in mind, it follows,

$$\alpha^* = \min_{\sigma \in \Sigma([C])} \min_{\ell \in [C]} \alpha_\ell(\sigma).$$

Therefore, in order to prove Equation (5) it is enough to prove that for any permutation $\sigma \in \Sigma([C])$ and any $\ell \in [C]$, Equation (5) holds for $\alpha_\ell(\sigma)$. Let us prove the property for $\sigma = \mathrm{I}_C$, the identity permutation. Notice this is done without loss of generality as the same argument will work for any other permutation $\sigma$. It follows,

$$\alpha_\ell = \frac{\mathrm{r}([\ell]) + \sum_{t \in [\ell]} \mathrm{r}(t) - \sum_{t \in [\ell]} \mathrm{r}(t)}{\sum_{t \in [\ell]} \mathrm{r}(t)}$$

$$= 1 - \frac{\sum_{t \in [\ell]} \mathrm{r}(t) - \mathrm{r}([\ell])}{\sum_{t \in [\ell]} \mathrm{r}(t)}$$

$$= 1 - \frac{\sum_{t\in[L]} \mathrm{r}(t) - \sum_{t=\ell+1}^{C} \mathrm{r}(t) - \mathrm{r}([\ell])}{\sum_{t\in[\ell]} \mathrm{r}(t)}$$

$$= 1 - \frac{C\mathrm{r} - \sum_{t=\ell+1}^{C} \mathrm{r}(t) - \mathrm{r}([\ell])}{\ell\mathrm{r}}$$

$$= 1 - \frac{C\mathrm{r} - \mathrm{r}([C]) - \sum_{t=\ell+1}^{C} \mathrm{r}(t) - \mathrm{r}([\ell]) + \mathrm{r}([C])}{\ell\mathrm{r}}$$

$$= 1 - \frac{C\mathrm{r} - \rho C\mathrm{r}}{\ell\mathrm{r}} + \frac{\sum_{t=\ell+1}^{C} \mathrm{r}(t) + \mathrm{r}([\ell]) - \mathrm{r}([C])}{\ell\mathrm{r}}$$

$$= 1 - \frac{C(1-\rho)}{\ell} + \frac{\sum_{t=\ell+1}^{C} \mathrm{r}(t) + \sum_{t=\ell+1}^{C} \mathrm{r}([t-1]) - \mathrm{r}([t])}{\ell\mathrm{r}}$$

$$= 1 - \frac{C(1-\rho)}{\ell} + \frac{\sum_{t=\ell+1}^{C}[\mathrm{r}(t) - \mathrm{r}([t]) + \mathrm{r}([t-1])]}{\ell\mathrm{r}}.$$

The numerator of the third term satisfies,

$$\sum_{t=\ell+1}^{C}[\mathrm{r}(t) - \mathrm{r}([t]) + \mathrm{r}([t-1])] \geq \max\{0, C(1-\rho) - (\ell-1)\}.$$

Indeed, the term is always non-negative as the rank function is submodular and non-negative, therefore, $\mathrm{r}([t]) = \mathrm{r}([t-1] \cup \{t\}) \leq \mathrm{r}([t-1]) + \mathrm{r}(t)$. The second lower bound comes from,

$$\sum_{t=\ell+1}^{C}[\mathrm{r}(t) - \mathrm{r}([t]) + \mathrm{r}([t-1])] = \sum_{t\in[C]} \mathrm{r}(t) - \mathrm{r}([C]) + \mathrm{r}([\ell]) - \sum_{t\in[\ell]} \mathrm{r}(t)$$

$$= C\mathrm{r}(1-\rho) + \mathrm{r}([\ell]) - \ell\mathrm{r}$$

$$\geq C\mathrm{r}(1-\rho) - (\ell-1)\mathrm{r},$$

where we have used that $\mathrm{r}([\ell]) \geq \mathrm{r}(\ell) = \mathrm{r}$. It follows,

$$\alpha_\ell \geq 1 - \frac{C(1-\rho)}{\ell} + \max\left\{0, \frac{C(1-\rho) - (\ell-1)}{\ell}\right\} = \max\left\{\frac{C(\rho-1)+\ell}{\ell}, \frac{1}{\ell}\right\}.$$

In particular, as the lower bound over $\alpha_\ell$ does not depend on the chosen permutation,

$$\alpha^* \geq \min_{\ell\in[C]} \max\left\{\frac{C(\rho-1)+\ell}{\ell}, \frac{1}{\ell}\right\},$$

whose minimum is attained at $\ell^* = (1-\rho)C + 1$. Remark the second upper bound is obtained by replacing $\ell^*$ in the previous inequality. Regarding the first upper bound, as $\ell$ must be an integer, the minimum is either reached at $\lfloor\ell^*\rfloor$ or $\lceil\ell^*\rceil$. It follows,

$$\alpha^* \geq \min\left\{\frac{C(\rho-1) + \lceil(1-\rho)C+1\rceil}{\lceil(1-\rho)C+1\rceil}, \frac{1}{\lfloor(1-\rho)C+1\rfloor}\right\} = \min\left\{\frac{C\rho - \lfloor C\rho\rfloor + 1}{C - \lfloor C\rho\rfloor + 1}, \frac{1}{C - \lfloor C\rho\rfloor}\right\},$$

which concludes the proof of the stated upper bound.

Regarding the tightness of the bound, we provide the example for transversal matroids. A similar construction can be done for graphic matroids by using the Hamiltonian cycle decomposition. Let $\rho \in [1/C, 1]$, $\rho \in \mathbb{Q}$, $\mathrm{r} \gg 1$, and denote $\ell^* := \lfloor C\rho\rfloor$ and $\mathrm{r}_1 = (C\rho - \ell^*)\mathrm{r}$. We take $\mathrm{r}$ such that $\mathrm{r}_1 \in \mathbb{N}$. Consider the following bipartite graph,

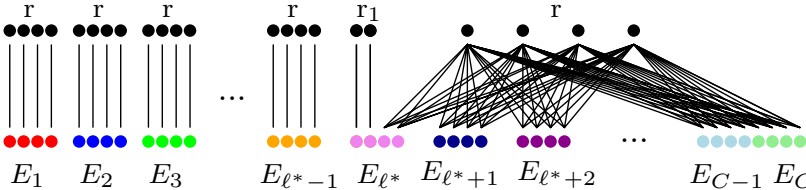

Figure 11: Tight bound example for PoF as stated in Proposition 4.6 for transversal matroids.

where each group $E_\ell$ has r elements. All groups $\ell \in \{1, ..., \ell^* - 1\}$ are independent and have $r(\ell) = r$. All groups $\ell \in \{\ell^* + 1, ..., C\}$ share resources and have $r(\ell) = r$. Finally, group $E_{\ell^*}$ is a *semi-independent* group, where $r_1$ agents are connected to $r_1$ resources and $r - r_1$ agents belong to the clique. We obtain $r(\ell^*) = r$ as well. Finally, remark $r([C]) = (\ell^* - 1)r + r_1 + r = C\rho r$.

We focus next on the maximum size opportunity fair allocation. Notice that, as all groups have the same rank, an allocation $x$ is opportunity fair if $x_\ell = x_k$ for all $\ell, k \in [C]$. Since all groups $\ell \in \{\ell^* + 1, ..., C\}$ share all their resources, the highest share than can be fairly allocated to them is,

$$x_\ell = \frac{r}{C - \ell^*}.$$

This allocation is feasible if and only if the remaining available resources to be allocated to $\ell^*$ are enough to fulfill its demand, i.e, if and only if,

$$r_1 \geq \frac{r}{C - \ell^*} \tag{6}$$

Moreover, remark that depending on whether Equation (6) holds or not, the maximum on the stated upper bound gets a different value,

$$\frac{r}{C - \lfloor C\rho \rfloor} \leq (L\rho - \lfloor C\rho \rfloor)r$$

$$\iff \frac{1}{C - \lfloor C\rho \rfloor} + 1 \leq C\rho - \lfloor C\rho \rfloor + 1$$

$$\iff \frac{C - \lfloor C\rho \rfloor + 1}{C - \lfloor C\rho \rfloor} \leq C\rho - \lfloor C\rho \rfloor + 1$$

$$\iff \frac{C - \lfloor C\rho \rfloor + 1}{C\rho - \lfloor C\rho \rfloor + 1} \leq C - \lfloor C\rho \rfloor.$$

Suppose Equation (6) holds. It follows the allocation is feasible and,

$$\|x\|_1 = \frac{Cr}{C - \ell^*} = \frac{Cr}{C - \lfloor C\rho \rfloor},$$

and the Price of Fairness is equal to,

$$\text{PoF} = \frac{C\rho r}{\frac{Cr}{C - \lfloor C\rho \rfloor}} = \rho(C - \lfloor C\rho \rfloor),$$

which is indeed equal to the upper bound. Suppose Equation (6) does not hold. In particular, the opportunity fair allocation must allocate some share of the r resources on the clique to $E_{\ell^*}$. Let $s \in (0, 1)$ denote the share. We obtain the following system,

$$sr + r_1 = \frac{(1 - s)r}{C - \ell^*},$$

whose solution is given by

$$s^* = \frac{1 - (C\rho - \ell^*)(C - \ell^*)}{C - \ell^* + 1}.$$

It follows the opportunity fair allocation $x$ has size,

$$\|x\|_1 = C \cdot \frac{(1 - s^*)r}{C - \ell^*} = \frac{C(C\rho - \ell^* + 1)r}{C - \ell^* + 1},$$

which yields,

$$\text{PoF} = \frac{C\rho r}{\frac{C(C\rho - \ell^* + 1)r}{C - \ell^* + 1}} = \rho \cdot \frac{C - \ell^* + 1}{C\rho - \ell^* + 1} = \rho \cdot \frac{C - \lfloor C\rho \rfloor + 1}{C\rho - \lfloor C\rho \rfloor + 1},$$

which corresponds to the stated upper bound when Equation (6) does not hold. $\qquad\square$

### E.9 Proof of Proposition 4.7

We recall that $(M_n(p))_{n\in\mathbb{N}}$ is a sequence of C-colored matroids over sets $(E^{(n)})$ such that $|E^{(n)}| = n$, randomly colored according to $p = (p_1, p_2, ..., p_C)$, and $(r_n)_{n\in\mathbb{N}}$ the associated sequence of rank functions. For each $c \in [C]$, suppose the following limit exists,

$$R(p_c) := \lim_{n\to\infty} \frac{\mathbb{E}_{p_c}[r_n(c)]}{n}, \tag{7}$$

and recall its natural extension to any subset $\Lambda \subseteq [C]$,

$$R\left(\sum_{c\in\Lambda} p_c\right) := \lim_{n\to\infty} \frac{1}{n} \cdot \mathbb{E}_{(p_c)_{c\in\Lambda}}\left[r_n\left(\bigcup_{c\in\Lambda} E_c\right)\right].$$

**Proposition 4.7.** If $R(1) > 0$, it follows

$$\text{PoF}(M_n(p)) \xrightarrow[n\to\infty]{\text{P}} \max_{\Lambda\subseteq[C]} \frac{R(1)}{\sum_{c\in[C]} R(p_c)} \cdot \frac{\sum_{c\in\Lambda} R(p_c)}{R(\sum_{c\in\Lambda} p_c)}, \tag{8}$$

where $\xrightarrow{\text{P}}$ denotes convergence in probability. Moreover, the function $R$ is such that $R(0) = 0$, $R$ is concave, non-decreasing, and 1-Lipschitz. Finally, for any such function $R$, there exists a double sequence of C-colored matroids $(M'_{n,m})$ such that, for $R'_m$ defined similarly to $R$ for the sequence $(M'_{n,m})_{n\in\mathbb{N}}$,

$$\|R - R'_m\|_\infty \xrightarrow[m\to\infty]{} 0.$$

*Proof.* Let $\mathcal{R} := \{f : [0,1] \to \mathbb{R} \mid f$ is a concave, non-decreasing, 1-Lipschitz function, and $f(0) = 0\}$. Remark that $\mathcal{R}$ is closed and convex. We divide the proof in several steps. First, we prove the function $R$ defined in Equation (7) belongs to $\mathcal{R}$. Second, we prove the PoF converges in probability to the stated limit. Third, we construct a double sequence of colored matroids $(M'_{n,m})_{n,m\in\mathbb{N}}$ such that $R$ is well approximated by $R_m$, where each $R_m$ is defined as in Equation (7) for $(M'_{n,m})_{n\in\mathbb{N}}$.

**1. Function $R$ belongs to $\mathcal{R}$.** Since $\mathcal{R}$ is closed, it is enough to prove that for each $n \in \mathbb{N}$, the mapping

$$Q_n : [0,1] \to \mathbb{R}_+$$
$$p_c \mapsto \frac{\mathbb{E}_{p_c}[r_n(c)]}{n}$$

belongs to $\mathcal{R}$. Clearly, $Q_n(0) = 0$. Regarding concavity and monotonicity, remark

$$\mathbb{E}_{p_c}[r_n(c)] = \sum_{S\subseteq E} r_n(S)\mathbb{P}(E_c = S) = \sum_{S\subseteq E} r_n(S)p_c^{|S|}(1-p_c)^{|E|-|S|},$$

which can be seen as the multi-linear extension of the rank function $r_n$ of $M_n$ evaluated at $(p_c, ..., p_c)$. It follows that $\mathbb{E}_{p_c}[r_n(c)]$ is a concave and non-decreasing function as $r_n$ is submodular [24]. Moreover, $Q_n$ is also concave and non-decreasing. Finally, remark,

$$\mathbb{E}_{p_c}[r_n(c)] \leq \mathbb{E}_{p_c}[|E_c|] = np_c,$$

where $n$ is the total number of agents in $E$. In particular, $Q_n$ is 1-Lipschitz[5].

**2. Convergence of PoF.** To prove the convergence of the PoF, remark first that $r_n(c)$ concentrates around its mean $\mathbb{E}_{p_c}[r_n(c)]$. Indeed, $r_n(c)$ is a function on the indicator variables $\mathbb{1}[e \in E_c]$ for $e \in E$, which are i.i.d. according to $\text{Ber}(p_c)$. In particular, $r_n(c)$ has a bounded difference of 1, as for any of the indicator variables that changes of value, the rank modifies at most in 1. The McDiarmid concentration inequality implies that,

$$\mathbb{P}\left(|r_n(c) - \mathbb{E}_{p_c}[r_n(c)]| \geq \sqrt{n\log(n)}\right) \leq \exp\left(\frac{-2n\log(n)}{n}\right) = \frac{1}{n^2}.$$

---

[5]Remark the function is defined over the interval $[0,1]$. Therefore, $Q_n$ is concave, increasing, $Q(0) = 0$, and $Q(x) \leq x$ for any $x \in [0,1]$, if and only if the function is 1-Lipschitz.

Added to the union bound, we obtain that,

$$\left| \sum_{c \in [C]} \frac{\mathrm{r}_n(c)}{n} - \sum_{c \in [C]} \frac{\mathbb{E}_{p_c}[\mathrm{r}_n(c)]}{n} \right| \xrightarrow[n \to \infty]{P} 0,$$

in other words,

$$\lim_{n \to \infty} \sum_{c \in [C]} \frac{\mathrm{r}_n(c)}{n} = \sum_{c \in [C]} R(p_c).$$

For any $c \in [C]$, notice that,

$$R(1) = \lim_{n \to \infty} \frac{\mathbb{E}_{p_c=1}[\mathrm{r}_n(c)]}{n} = \lim_{n \to \infty} \frac{\mathbb{E}[\mathrm{r}_n(E)]}{n} = \lim_{n \to \infty} \frac{\mathrm{r}_n(E)}{n}.$$

Next, since $R$ is concave,

$$\lim_{n \to \infty} \frac{\mathbb{E}_{p_c}[\mathrm{r}_n(c)]}{n} = R(p_c) = R(p_c \cdot 1 + (1 - p_c) \cdot 0) \geq p_c R(1) + (1 - p_c) R(0) = p_c \lim_{n \to \infty} \frac{\mathrm{r}_n(E)}{n}.$$

Since $R(1) > 0$, we obtain that both $\mathrm{r}_n(E) = \Omega(n)$ and $\mathbb{E}_{p_c}[\mathrm{r}_n(c)] = \Omega(n)$. Putting all together, we conclude the following,

$$\mathrm{PoF}(M_n) = \max_{\Lambda \subseteq [C]} \frac{\mathrm{r}_n([C])}{\sum_{c \in [C]} \mathrm{r}_n(c)} \cdot \frac{\sum_{c \in \Lambda} \mathrm{r}_n(c)}{\mathrm{r}_n(\Lambda)} \xrightarrow[n \to \infty]{} \max_{\Lambda \subseteq [C]} \frac{R(1)}{\sum_{c \in [C]} R(p_c)} \cdot \frac{\sum_{c \in \Lambda} R(p_c)}{R(\sum_{c \in \Lambda} p_c)},$$

where we used the assumption that $R$ exists and that $R(p_c) > 0$ for all $c$.

**3. Approximation result**. To approximate the functions in $\mathcal{R}$, we will construct a family of matroids able to produce a family of piece-wise functions $f \in \mathcal{R}$ whose convex hull i dense on the set $\mathcal{R}$. Let $0 \leq b \leq a \leq 1$ be two real values and $n \in \mathbb{N}$, such that $an, bn$, and $(1 - a)n$ are integer values. Consider the following graph containing a complete bipartite graph with sides of sizes $an$ and $bn$, respectively, and $(1 - a)n$ isolated vertices, as in the figure below,

$$an \qquad\qquad\qquad (1-a)n$$

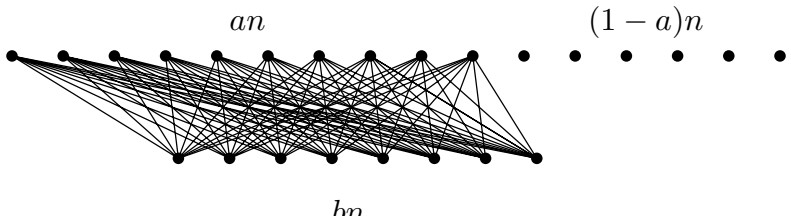

$$bn$$

Let $M_n$ be the associated transversal matroid. Given a random coloring according to a vector $p = (p_1, ..., p_C)$, notice that, for $n$ large enough,

$$\mathbb{E}_{p_c}[\mathrm{r}_n(c)] = \min\{a p_c n, bn\}.$$

For this sequence of matroids, it follows that,

$$R(p_c) = \lim_{n \to \infty} \frac{\mathbb{E}_{p_c}[\mathrm{r}_n(c)]}{n} = \min\{a p_c, b\}.$$

We denote

$$\mathcal{T} := \{f : [0, 1] \to \mathbb{R} \mid \exists a, b \in \mathbb{R}_+, f(t) = \min\{at, b\}, \forall t \in [0, 1]\}.$$

In particular, all functions in $\mathcal{T}$ can be obtained by the previous construction. Consider next the set,

$$\mathcal{H} := \{f : [0, 1] \to \mathbb{R} \mid f \text{ is piece-wise linear, concave, non-decreasing, 1-Lipschitz, and } f(0) = 0\}.$$

We claim that any function in $\mathcal{H}$ can be obtained as convex combinations of functions within $\mathcal{T}$. Indeed, for $f \in \mathcal{H}$ consisting in two pieces of value $a$ and then $b \leq a$, i.e, such that there exists $t^* \in [0, 1]$,

$$f(t) = \begin{cases} at & t \leq t^*, \\ b(t - t^*) + at^* & t \geq t^*, \end{cases}$$

$$f_1 : \ [0,1] \to \mathbb{R}, \quad f_2 : \ [0,1] \to \mathbb{R}$$
$$t \mapsto at \qquad\qquad t \mapsto \max(at, at^*)$$

it is enough to take

as $f \equiv \frac{b}{a} f_1 + (1 - \frac{b}{a}) f_2$. For the rest of functions within $\mathcal{H}$, the construction is done inductively. Consider $f \in \mathcal{H}$ to be $(m+1)$-piece-wise linear, for $m \geq 2$. Let $0 \leq c \leq b \leq a \leq 1$ be the last three linear slopes of $f$ with respective changes at $t_1 \leq t_2$, as illustrated in Figure 12.

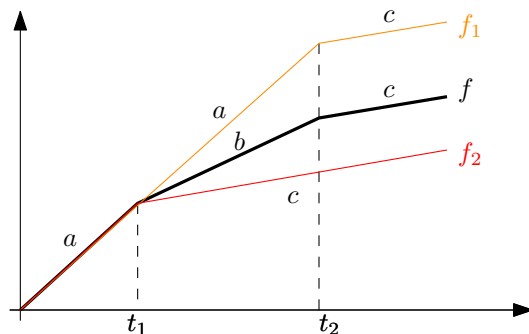

Figure 12: Example piece-wise function

Consider next,

$$f_1(t) := \left\{ \begin{array}{ll} f(t) & t \leq t_1 \\ f(t_1) + at & t \in [t_1, t_2] \\ f(t_1) + a(t_2 - t_1) + ct & t \geq t_2 \end{array} \right. \quad \text{and} \quad f_2(t) := \left\{ \begin{array}{ll} f(t) & t \leq t_1 \\ f(t_1) + ct & t \geq t_1 \end{array} \right.$$

Remark both $f_1$ and $f_2$ are $m$-piece-wise linear. It is not hard to check that

$$f \equiv \left( \frac{b-c}{a-c} \right) f_1 + \left( 1 - \left( \frac{b-c}{a-c} \right) \right) f_2.$$

We conclude the proof by showing that $\mathcal{H}$ is dense in $\mathcal{R}$. Let $R \in \mathcal{R}$ be fixed. For $m \in \mathbb{N}$, divide the interval $[0,1]$ in $m$ pieces $\{0, \frac{1}{m}, \frac{2}{m}, ..., \frac{m-1}{m}, 1\}$, and define the $m$-piece-wise linear function that interpolates $R$, as it follows,

$$f(t) = R(i) + t \left( R\left( \frac{i+1}{m} \right) - R\left( \frac{i}{m} \right) \right), \ \text{for } t \in \left[ \frac{i}{m}, \frac{i+1}{m} \right], \ \text{for } i \in \{0, 1, ..., m\}.$$

For $t \in [\frac{i}{m}, \frac{i+1}{m}]$, by monotonicity of $R$ and $f$, it follows,

$$|R(t) - f(t)| \leq \max \left\{ R\left( \frac{i+1}{m} \right) - f\left( \frac{i}{m} \right); f\left( \frac{i+1}{m} \right) - R\left( \frac{i}{m} \right) \right\}$$
$$= R\left( \frac{i+1}{m} \right) - R\left( \frac{i}{m} \right)$$
$$\leq \frac{1}{m} \xrightarrow[m \to \infty]{} 0,$$

where the last inequality comes from the fact that $R$ is 1-Lipschitz. In particular, $\|R - f\|_\infty \to 0$. $\quad \square$

### E.10  Proof of Theorem 4.8

**Theorem 4.8**. Let $\pi \in [0,1]$ be fixed. Consider the sets,

$$\mathcal{R} := \big\{ f : [0,1] \to \mathbb{R} \mid f \text{ is a concave, non-decreasing, 1-Lipschitz function, and } f(0) = 0 \big\},$$
$$\Delta_\pi^C := \big\{ p \in \Delta^C \mid \max_{c \in [C]} p_c = \pi \big\}.$$

It follows,

$$\max_{p\in\Delta_\pi^C}\max_{R\in\mathcal{R}}\max_{\Lambda\subseteq[C]}\frac{R(1)}{\sum_{c\in[C]}R(p_c)}\cdot\frac{\sum_{c\in\Lambda}R(p_c)}{R(\sum_{c\in\Lambda}p_c)}=\max_{\lambda\in[C]}\psi_\lambda\left(\frac{1-(C-\lambda)\pi}{C}\right)\leq C-\frac{1}{\pi}, \quad (9)$$

where $\psi_\lambda:[-\lambda,\frac{1}{C}]\to\mathbb{R}$, for each $\lambda\in[C]$, is given by,

$$\psi_\lambda(q)=\begin{cases}\lambda & q\in[-\lambda,0]\,,\\\frac{\lambda}{(\lambda Cq-1)^2}\cdot\left(1+q(1-2\lambda)+C(\lambda-2+\lambda q)q-2\sqrt{(\lambda-1)(C-1)(1-Cq)(1-\lambda q)q}\right) & q\in\left(0,\frac{(\lambda-1)}{\lambda(C-1)}\right],\\1 & q\in\left(\frac{(\lambda-1)}{\lambda(C-1)},\frac{1}{C}\right].\end{cases}$$
$$(10)$$

The proof of Theorem 4.8 consists of constructing an optimal solution of the triple optimization problem in Equation (9) by starting from an instance $(p_0,\Lambda_0,R_0)$ and iteratively modifying $p$, $\Lambda$, and $R$. Before giving the formal proof, we show some useful technical lemmas. We define

$$F:\Delta^C\times\mathcal{R}\times 2^{[C]}\longrightarrow[1,\infty)$$

$$(p,R,\Lambda)\longrightarrow F(p,R,\Lambda):=\frac{R(1)}{\sum_{c\in[C]}R(p_c)}\cdot\frac{\sum_{c\in\Lambda}R(p_c)}{R(\sum_{c\in\Lambda}p_c)}.$$

Remark that whenever $|\Lambda|\in\{1,C\}$, $F(p,R,\Lambda)=1$, for any $p,R\in\Delta^C\times\mathcal{R}$. Indeed,

$$F(p,R,\{\bar{c}\})=\frac{R(1)}{\sum_{c\in[C]}R(p_c)}\leq 1,$$

where the inequality comes from the concavity of $R$ and the fact that $\sum_{c\in[C]}p_c=1$, so $\sum_{c\in[C]}R(p_c)\leq R(\sum_{c\in[C]}p_c)$. Similarly,

$$F(p,R,[C])=\frac{R(1)}{R(\sum_{c\in[C]}p_c)}=1.$$

Therefore, from now on, **we suppose $1<|\Lambda|<C$**. The function $F$ is invariant to scaling $R$ by non-null constants, i.e., $F(p,R,\Lambda)=F(p,\alpha R,\Lambda)$ for any $\alpha\neq 0$. In addition, $F$ evaluates $R$ at $C+2$ points: $(p_c)_{c\in[C]}$, $\sum_{c\in\Lambda}p_c$, and 1. Since,

$$F(p,R,\Lambda)=\frac{R(1)}{R(\sum_{c\in\Lambda}p_c)}\left(1-\frac{\sum_{c\in[C]\setminus\Lambda}R(p_c)}{\sum_{c\in[C]}R(p_c)}\right),$$

$F$ is decreasing on $R(\sum_{c\in\Lambda}p_c)$ and $(R(p_c))_{c\in[C]\setminus\Lambda}$ and increasing on $R(1)$ and $(R(p_c))_{c\in\Lambda}$. Figure 13 illustrates a function $R\in\mathcal{R}$ for $C=5$ and $\Lambda=\{1,2,4\}$, with the red dots indicating the values where $F$ is decreasing, and the blue dots those where $F$ is increasing.

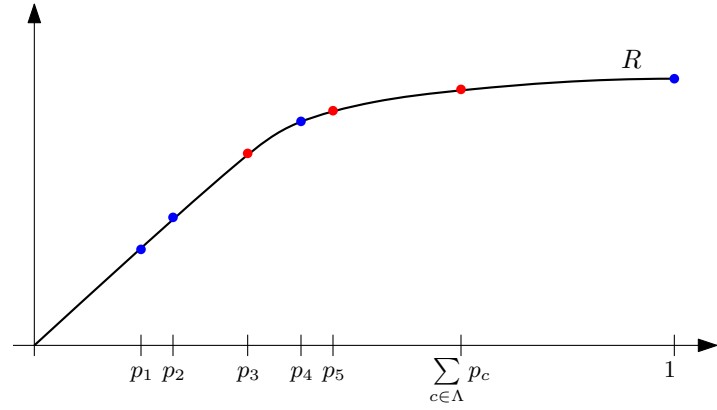

Figure 13: Increasing (blue) and decreasing (red) points for function $F$

The construction in the proof of Theorem 4.8 will be done by playing with both: the position (over the horizontal axis) of the red and blue dots and their values.

**Lemma E.1.** *Given $(p, R, \Lambda) \in \Delta^C \times \mathcal{R} \times 2^{[C]}$, we can always construct $R' \in \mathcal{R}$ such that either $F(p, R', \Lambda) > F(p, R, \Lambda)$ or $R' = R$.*

*Proof.* Given $(p, R, \Lambda) \in \Delta^C \times \mathcal{R} \times 2^{[C]}$, it is enough with picking $R' \in \mathcal{R}$ satisfying,

$$R'\left(\sum_{c \in \Lambda} p_c\right) \leq R\left(\sum_{c \in \Lambda} p_c\right)$$

$$R'(p_c) \leq R(p_c), \forall c \in [C] \setminus \Lambda$$

$$R(1) \leq R'(1)$$

$$R(p_c) \leq R'(p_c), \forall c \in \Lambda.$$

For example, suppose $C = 3, 0 < p_1 < p_2 < p_3 < p_2 + p_3 < 1$, and $\Lambda = \{2, 3\}$. Starting from $R$ we can take $R' \in \mathcal{R}$ such that

$$R'(x) = \begin{cases} x \cdot \frac{R(p_2)}{p_2} & x \in [0, p_2] \\ R(x) & x \in [p_2, p_3] \\ R(p_3) + (x - p_3) \cdot \frac{R(1) - R(p_3)}{1 - p_3} & x \in [p_3, 1] \end{cases}$$

as illustrated in Figure 14,

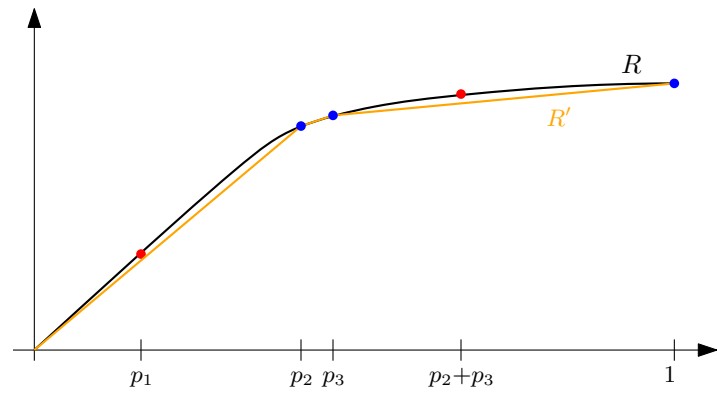

Figure 14: Function $R'$

Remark that

$$R'(p_2 + p_3) = R(p_3) + p_2 \cdot \frac{R(1) - R(p_3)}{1 - p_3}$$

$$= \frac{p_2}{1 - p_3} \cdot R(1) + \left(1 - \frac{p_2}{1 - p_3}\right) \cdot R(p_3)$$

$$\leq R\left(\frac{p_2}{1 - p_3} + \left(1 - \frac{p_2}{1 - p_3}\right) \cdot p_3\right)$$

$$= R\left(\frac{1}{1 - p_3} \cdot p_2(1 - p_3) + p_3\right) = R(p_2 + p_3),$$

where the inequality comes from $R$'s concavity. Similarly,

$$R'(p_1) = \frac{p_1}{p_2} \cdot R(p_2)$$

$$= \frac{p_1}{p_2} \cdot R(p_2) + \left(1 - \frac{p_1}{p_2}\right) R(0)$$

$$\leq R\left(\frac{p_1}{p_2} \cdot p_2 + \left(1 - \frac{p_1}{p_2}\right) \cdot 0\right) = R(p_1),$$

where we have used $R$'s concavity and that $R(0) = 0$. Finally, remark $R(1) = R'(1)$ and $R'(p_c) = R(p_c)$ for $c \in \Lambda$. $\qquad \square$

**Lemma E.2.** *Let $\pi \in [0,1]$ be fixed, $(p, R, \Lambda) \in \Delta_\pi^C \times \mathcal{R} \times 2^{[C]}$, and $c^* = \mathrm{argmax}_{c \in \Lambda} \, p_c$ (if several $c^*$ exists, pick one at random). Consider $p', p'' \in \Delta_\pi^C$ given by,*

$$\forall c \in [C], p'_c = \left\{ \begin{array}{ll} p_c & c \in [C] \setminus \Lambda \text{ or } c = c^* \\ \frac{1}{|\Lambda|-1} \sum_{c \in \Lambda \setminus \{c^*\}} p_c & c \in \Lambda \setminus \{c^*\}, \end{array} \right.$$

$$\forall c \in [C], p''_c = \left\{ \begin{array}{ll} p_c & c \in [C] \setminus \Lambda \\ \frac{1}{|\Lambda|} \sum_{c \in \Lambda} p_c & c \in \Lambda. \end{array} \right.$$

*It follows $F(p, R, \Lambda) \leq F(p', R, \Lambda)$ and $F(p, R, \Lambda) \leq F(p'', R, \Lambda)$.*

To prove Lemma E.2 we introduce the following definition.

**Definition E.3.** For $x \in \mathbb{R}_+^C$ a vector, we denote $x_{(c)}$ to its $c$-th highest entry. Given $x, y \in \mathbb{R}_+^C$, we say that $x$ **majorizes** $y$ if

$$\sum_{c=1}^\lambda x_{(c)} \geq \sum_{c=1}^\lambda y_{(c)}, \text{ for all } \lambda \in [C], \text{ and } \sum_{c=1}^C x_c = \sum_{c=1}^C y_c.$$

In addition, we state **Kamarata's inequality**: Let $x, y \in \mathbb{R}^C$ be two vectors such that $x$ majorizes $y$. For any concave function $f$, it follows,

$$\sum_{c \in [C]} f(x_c) \leq \sum_{c \in [C]} f(y_c).$$

*Proof of Lemma E.2.* We prove the stated result for $p'$. For $p''$ the argument is analogous. Recall that $F$ is increasing on $\sum_{c \in \Lambda} R(p_c)$. We prove that $(p_c)_{c \in \Lambda}$ majorizes $(p'_c)_{c \in \Lambda}$ and conclude by using Karamata's inequality over $R$. First, remark

$$\sum_{c \in \Lambda} p'_c = p_{c^*} + \sum_{c \in \Lambda \setminus \{c^*\}} \frac{1}{|\Lambda|-1} \sum_{c \in \Lambda \setminus \{c^*\}} p_c = p_c^* + \sum_{c \in \Lambda \setminus \{c^*\}} p_c = \sum_{c \in \Lambda} p_c.$$

Regarding the inequality, assume without loss of generality that $\Lambda = \{1, ..., m\}$ and $p_1 \geq p_2 \geq ... \geq p_m$. In particular, notice $p'_1 \geq p_1$ (as they are equal). For any $\lambda \in \{2, ..., m-1\}$, it follows,

$$\begin{aligned} \sum_{c=1}^\lambda p'_c &= p_1 + \sum_{c=2}^\lambda \frac{1}{m-1} \sum_{c=2}^m p_c \\ &= p_1 + \frac{\lambda-1}{m-1} \sum_{c=2}^m p_c \\ &= p_1 + \frac{\lambda-1}{m-1} \sum_{c=2}^\lambda p_c + \frac{\lambda-1}{m-1} \sum_{c=\lambda+1}^m p_c \\ &= p_1 + \sum_{c=2}^\lambda p_c - \frac{m-\lambda}{m-1} \sum_{c=2}^\lambda p_c + \frac{\lambda-1}{m-1} \sum_{c=\lambda+1}^m p_c \\ &\leq p_1 + \sum_{c=2}^\lambda p_c = \sum_{c=1}^\lambda p_c, \end{aligned}$$

where the last inequality comes from the fact that,

$$(m-\lambda)(\lambda-1)p_\lambda \leq (m-\lambda) \cdot \sum_{c=2}^\lambda p_c \text{ and } (\lambda-1) \cdot \sum_{c=\lambda+1}^m p_c \leq (\lambda-1)(m-\lambda)p_{\lambda+1},$$

and therefore,

$$\frac{\lambda-1}{m-1} \sum_{c=\lambda+1}^m p_c - \frac{m-\lambda}{m-1} \sum_{c=2}^\lambda p_c \leq \frac{(m-\lambda)(\lambda-1)}{m-1} \cdot (p_{\lambda+1} - p_\lambda) \leq 0,$$

for any $\lambda \in \{2, ..., m-1\}$. $\qquad\square$

Lemma E.2 allows to replace all elements $p_c \in \Lambda$ by one single value equal to their mean. In particular, Figure 13 becomes

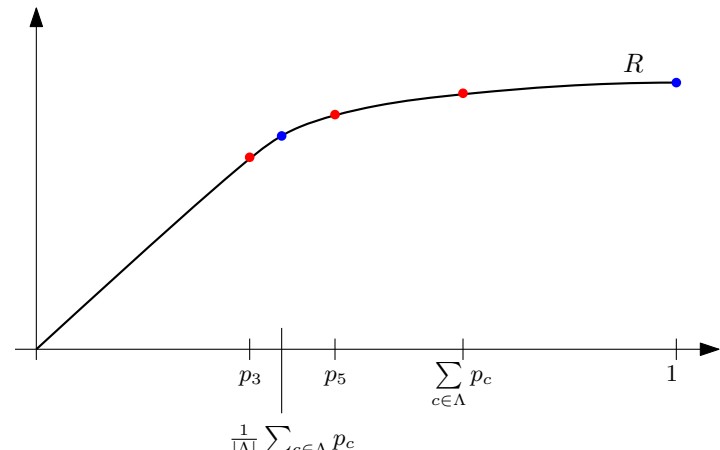

The main issue with the uniformization of the probabilities within $\Lambda$ is the fact that any transformation done to the vector $p \in \Delta_\pi^C$ must produce a vector within $\Delta_\pi^C$, i.e., the maximum-value must remain unchanged (although it could eventually change of index). The following Lemma shows that for any instance $(p, R, \Lambda) \in \Delta_\pi^C \times \mathcal{R} \times 2^{[C]}$, we can always modify $R$ and $\Lambda$ such that the maximum-value entry of $p$ stays outside of $\Lambda$, with a transformation that does not decrease the value of $F(p, R, \Lambda)$.

**Lemma E.4.** *Let $\pi \in [0, 1]$ be fixed, $(p, R, \Lambda) \in \Delta_\pi^C \times \mathcal{R} \times 2^{[C]}$, and $c^* = \mathrm{argmax}_{c \in [C]} \, p_c$ (if several $c^*$ exists, pick one at random), i.e., $p_{c^*} = \pi$. Then, we can always construct $(R', \Lambda') \in \mathcal{R} \times 2^{[C]}$ such that $c^* \notin \Lambda'$ and $F(p, R, \Lambda) \leq F(p, R', \Lambda')$.*

*Proof.* Suppose that $c^* \in \Lambda$. Apply the partial uniformization technique to $p$ of Lemma E.2 leaving $p_{c^*}$ unchanged. Denote

$$q := \frac{1}{|\Lambda| - 1} \sum_{c \in \Lambda \setminus \{c^*\}} p_c.$$

Since $p_{c^*} \in \Lambda$, remark $F$ is increasing at $R(1), R(q)$, and $R(\pi)$. Notice that $0 < q < \pi < \sum_{c \in \Lambda} p_c < 1$. Apply Lemma E.1 and replace $R$ by

$$R'(x) = \begin{cases} R(x) & x \in [0, q] \\ R(q) + (x - q)\frac{R(\pi) - R(q)}{\pi - q} & x \in [q, \pi] \\ R(\pi) + (x - \pi)\frac{R(1) - R(\pi)}{1 - \pi} & x \in [\pi, 1], \end{cases}$$

as illustrated in Figure 15, for $C = 5$, $\Lambda = \{1, 2, 4\}$, and $p_5 = \pi$. Remark that for $x \in [q, \pi]$ no value $R(x)$ is considered on $F$, which in particular allows to replace $R$ by the linear segment between the points $(q, R(q))$ and $(\pi, R(\pi))$.

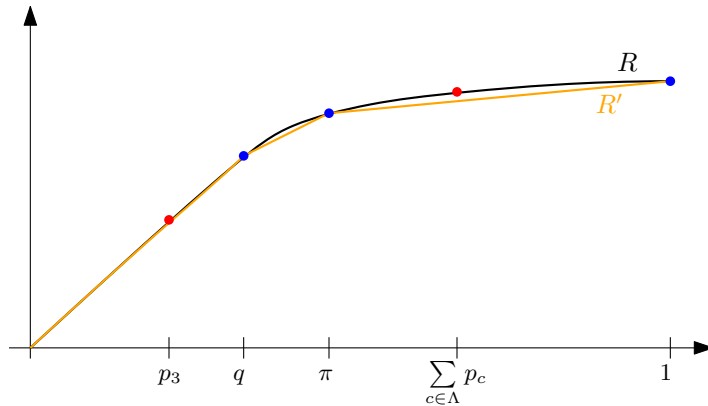

Figure 15: Function $R'$ Lemma E.4

Next, we show we can find $\Lambda' \subseteq [C] \setminus \{c^*\}$ and $R'' \in \mathcal{R}$ starting from $R'$ such that $F(p, R', \Lambda) \leq F(p, R'', \Lambda')$. Given $\varepsilon > 0$, consider

$$R'_\varepsilon(x) := \begin{cases} R'(x) & x \in [0, \pi] \\ R'(\pi) + (x - \pi)\left(\frac{R(1) - R(\pi)}{1 - \pi} + \varepsilon\right) & x \in [\pi, 1]. \end{cases}$$

Let

$$\varepsilon^* = \operatorname{argmax}\{\varepsilon : R'_\varepsilon \in \mathcal{R} \text{ and } F(p, R'_\varepsilon, \Lambda) \geq F(p, R', \Lambda)\},$$

and set $R'' = R'_{\varepsilon^*}$. We claim that

$$\frac{R(1) - R(\pi)}{1 - \pi} + \varepsilon^* = \frac{R(\pi) - R(q)}{\pi - q},$$

i.e., the segment between $(q, R''(q))$ and $(\pi, R''(\pi))$ has the same slope as the one between $(\pi, R''(\pi))$ and $(1, R''(1))$, as illustrated in Figure 16, for $C = 5$, $\Lambda = \{1, 2, 4\}$, and $p_5 = \pi$. Clearly, $R'_{\varepsilon^*}$ belongs to $\mathcal{R}$. Regarding the increase on the value of the function $F$, we show that the

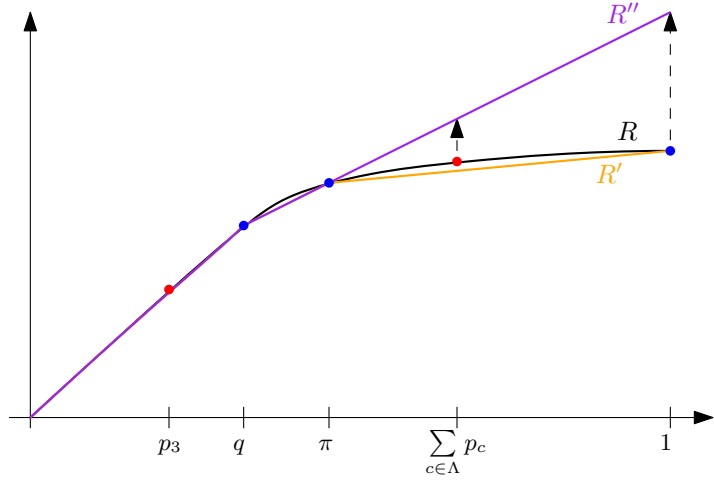

Figure 16: Function $R''$

for any $\varepsilon > 0$,

$$R'_\varepsilon\left(\sum_{c \in \Lambda} p_c\right) - R'\left(\sum_{c \in \Lambda} p_c\right) \leq R'_\varepsilon(1) - R'(1),$$

i.e, that the increase of the blue dot in Figure 16 is larger than the increase of the red dot. It follows,

$$\frac{d}{d\varepsilon}\left[\frac{R'_\varepsilon(1)}{R'_\varepsilon\left(\sum_{c\in\Lambda}p_c\right)}\right] = \frac{d}{d\varepsilon}\left[\frac{R'(\pi)+(1-\pi)\left(\frac{R(1)-R(\pi)}{1-\pi}+\varepsilon\right)}{R'(\pi)+\left(\sum_{c\in\Lambda}p_c-\pi\right)\left(\frac{R(1)-R(\pi)}{1-\pi}+\varepsilon\right)}\right]$$

$$= \frac{R'(\pi)(1-\sum_{c\in\Lambda}p_c)}{\left(R'(\pi)+\left(\sum_{c\in\Lambda}p_c-\pi\right)\left(\frac{R(1)-R(\pi)}{1-\pi}+\varepsilon\right)\right)^2},$$

which is always non-negative. In particular, $R''$ is rewritten as

$$R''(x) := \begin{cases} R(x) & x\in[0,q] \\ R(q)+(x-q)\left(\frac{R(\pi)-R(q)}{\pi-q}\right) & x\in[q,1]. \end{cases} \tag{11}$$

To ease the notation, we drop the $''$ from $R''$ and denote $\alpha = {(R(\pi)-R(q))}/{(\pi-q)}$. Finally, we construct $\Lambda' \subseteq [C]\setminus\{c^*\}$ such that $F(p,R,\Lambda) \leq F(p,R,\Lambda')$. The analysis is split depending on whether a value $p_{\bar{c}}$ with $\bar{c}\in[C]\setminus\Lambda$ (a red dot) lies between $q$ and $\pi$ or not. Suppose it does. We claim that considering $\Lambda' := \Lambda\setminus\{c^*\}\cup\{\bar{c}\}$ we obtain the stated result. Indeed, although swapping the elements should decrease the value of the function $F$ (as we obtain a higher-value red dot and a lower-value blue dot), the effect is compensated by the fact that $\sum_{c\in\Lambda'}p_c < \sum_{c\in\Lambda}p_c$. Figure 17 illustrates the swapping.

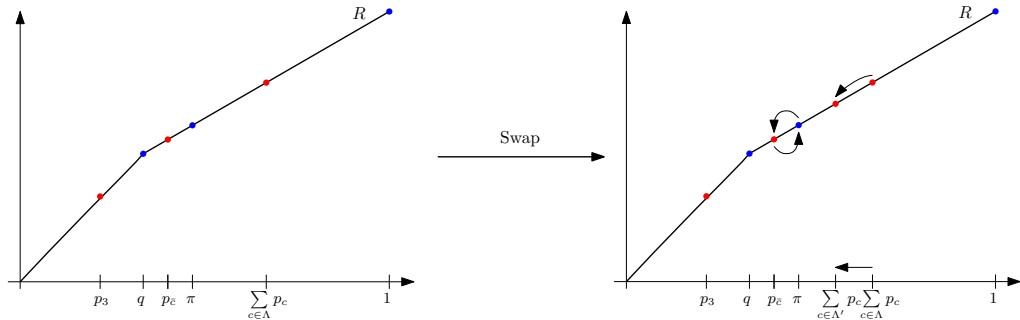

Figure 17: Swapping $p_{\bar{c}}$ and $\pi$ within $\Lambda$

To prove that $F(p,R,\Lambda) \leq F(p,R,\Lambda')$, we check that

$$\frac{\sum_{c\in\Lambda}R(p_c)}{R(\sum_{c\in\Lambda}p_c)} \leq \frac{\sum_{c\in\Lambda'}R(p_c)}{R(\sum_{c\in\Lambda'}p_c)}. \tag{12}$$

For $z\in[p_{\bar{c}},\pi]$, consider,

$$Q(z) := \frac{(|\Lambda|-1)R(q)+R(\pi-p_{\bar{c}}+z)}{R((|\Lambda|-1)q+\pi-p_{\bar{c}}+z)} = \frac{|\Lambda|R(q)+\alpha(\pi-p_{\bar{c}}+z-q)}{R(q)+\alpha((|\Lambda|-2)q+\pi-p_{\bar{c}}+z)}$$

where the last equality comes from using $R$'s definition (11). In particular Equation (12) holds if and only if $Q(\pi) \leq Q(p_{\bar{c}})$. Notice,

$$\frac{d}{dz}Q(z) = \frac{(R(q)+\alpha((|\Lambda|-2)q+\pi-p_{\bar{c}}+z))\alpha - (|\Lambda|R(q)+\alpha(\pi-p_{\bar{c}}+z-q))\alpha}{\left[R(q)+\alpha((|\Lambda|-2)q+\pi-p_{\bar{c}}+z)\right]^2}$$

$$= \frac{\alpha(|\Lambda|-1)(\alpha q - R(q))}{\left[R(q)+\alpha((|\Lambda|-2)q+\pi-p_{\bar{c}}+z)\right]^2} \leq 0,$$

where we have used that $R(q) \geq \alpha q$, which holds as $\alpha \leq 1$ is the slope of the last piece-wise part of the function $R$, which extended up to the origin remains positive, in particular implying that the image of 0 (given by $R(q) - \alpha q$) is at least 0. We conclude $Q(z)$ is decreasing over $[p_{\bar{c}},\pi]$, concluding that Equation (12) holds.

To finish the proof, suppose that such as $p_{\bar{c}}$ did not exist, as in Figure 16. Keep increasing the slope of the last pice-wise linear function until achieving the slop between $q$ and the closest red dot placed at the left of $q$, namely $p_{\underline{c}}$, as in Figure 18 and set $\Lambda' := \Lambda \setminus \{\bar{c}\} \cup \{\underline{c}\}$.

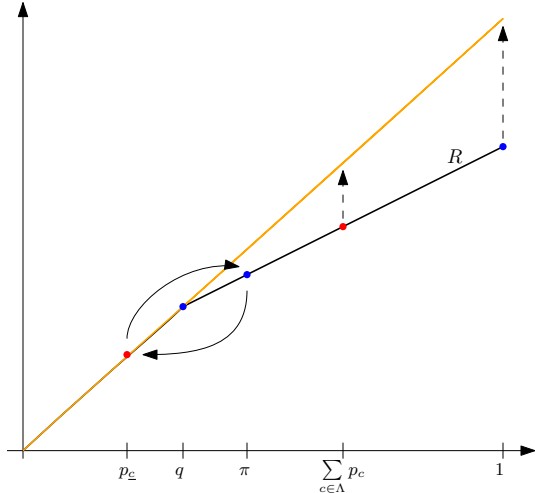

Figure 18: Final construction $\Lambda'$

As in the previous cases, it can be proved that $F(p, R, \Lambda) \leq F(p, R, \Lambda')$. We omit the proof. $\qquad\square$

**Lemma E.5.** *Let $\pi \in [0,1]$ be fixed and $(p, R, \Lambda) \in \Delta_\pi^C \times \mathcal{R} \times 2^{[C]}$. Define*

$$\Gamma := \left\{ k \in [C] \setminus \Lambda : p_k \leq \frac{1}{|\Lambda|} \sum_{c \in \Lambda} p_c \right\}.$$

*There exists $(p', R') \in \Delta_\pi^C \times \mathcal{R}$ such that $F(p, R, \Lambda) \leq F(p', R', \Lambda \cup \Gamma)$.*

*Proof.* Apply Lemmas E.2 and E.4 so the entry of $p$ of value $\pi$ is not included in $\Lambda$ and for any $c \in \Lambda$, $p_c = q := \frac{1}{|\Lambda|} \sum_{c \in \Lambda} p_c$. In particular, the only values where $F$ is increasing are $R(q)$ and $R(1)$. Define $\Gamma := p_c < q\}$. It follows that $F$ is decreasing at $(R(p_c))_{c \in \Gamma}$. Replace $R$ by

$$R(x) = \begin{cases} x \frac{R(q)}{q} & x \in [0, q] \\ \\ R(x) & x \in [q, \pi]. \end{cases}$$

Moreover, since $F(p, \alpha R, \Lambda) = F(p, R, \Lambda)$ for any $\alpha \neq 0$, redefine $R \equiv \frac{q}{R(q)} R$. Finally, since for any $q < p_c < 1$ the function $F$ is decreasing on $R(p_c)$, replace $R$ by,

$$R(x) = \begin{cases} x & x \in [0, q] \\ \\ q + (x - q) \cdot \frac{R(1) - q}{1 - q} & x \in [q, \pi], \end{cases}$$

where we have used that $R(x) = x$ for any $x \leq q$ because of the previous scaling. The resulting function is illustrated in Figure 19 for $\Gamma = \{1, 2, 3\}$.

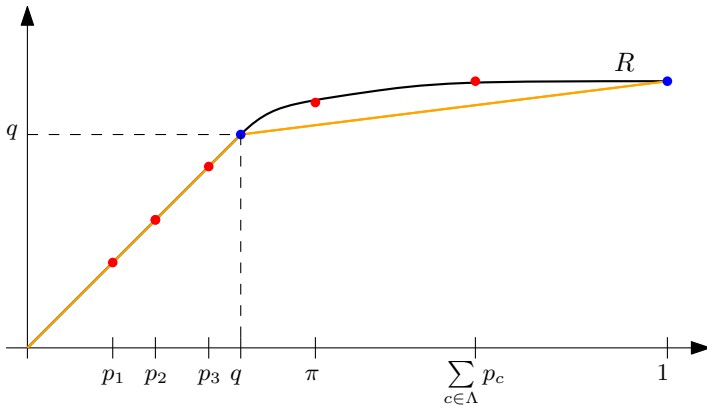

Figure 19: Function $R$ Lemma E.5

Finally, we prove that $F(p, R, \Lambda) \leq F(p, R, \Lambda \cup \Gamma)$. For $z \in [0, q]$, consider

$$Q(z) := \frac{|\Lambda| R(q) + R(z)}{R(|\Lambda| q + z)} = \frac{|\Lambda| q + z}{q + (|\Lambda| q + z - q)\alpha},$$

where $\alpha = \frac{R(1)-q}{1-q}$. Remark the previous claim holds if and only if $Q(0) \leq Q(z)$, i.e., adding elements from $\Gamma$ to $\Lambda$ increases the part of $F$ that depends on $\Lambda$. We obtain,

$$\begin{aligned}
\frac{d}{dz}Q(z) &= \frac{d}{dz}\left[\frac{|\Lambda| q + z}{q + (|\Lambda| q + z - q)\alpha}\right] \\
&= \frac{(q + (|\Lambda| q + z - q)\alpha) - (|\Lambda| q + z)\alpha}{\left(q + (|\Lambda| q + z - q)\alpha\right)^2} \\
&= \frac{q(1 - \alpha)}{\left(q + (|\Lambda| q + z - q)\alpha\right)^2},
\end{aligned}$$

which is always positive. We conclude the proof. $\qquad\square$

**Lemma E.6.** *Let $\pi \in [0, 1]$ be fixed and $(p, R, \Lambda) \in \Delta_\pi^C \times \mathcal{R} \times 2^{[C]}$. Then, there always exists $(p', R', \Lambda') \in \Delta_\pi^C \times \mathcal{R} \times 2^{[C]}$ such that,*

$$F(p', R', \Lambda') = \frac{\lambda}{\alpha(\lambda - 1) + 1} \cdot \frac{\alpha + (1 - \alpha)q}{\alpha + (1 - \alpha)Cq} =: \hat{F}(q),$$

*where $\lambda = |\Lambda'|$, $q = \frac{1}{\lambda}\sum_{c \in \Lambda'} p_c$, $\alpha = \frac{(R(1)-q)}{(1-q)}$, and $F(p, R, \Lambda) \leq F(p', R', \Lambda')$. In particular,*

$$\underset{q \in [0,1]}{\mathrm{argmax}}\, \hat{F}(q) = \left\{ \begin{array}{c} 0 \\ \frac{1 - (C - \lambda)\pi}{\lambda} \end{array} \right.$$

*Proof.* Let $\pi \in [0, 1]$ be fixed and $(p, R, \Lambda) \in \Delta_\pi^C \times \mathcal{R} \times 2^{[C]}$. Apply Lemmas E.1, E.2, E.4 and E.5 to construct $(p', R', \Lambda') \in \Delta_\pi^C \times \mathcal{R} \times 2^{[C]}$ such that,

$$F(p, R, \Lambda) \leq F(p', R', \Lambda')$$
$$\text{for any } c \in \Lambda', p'_c = q := \frac{1}{\lambda}\sum_{c \in \Lambda'} p_c$$
$$\text{for any } c \in [C] \setminus \Lambda, p'_c \geq q,$$
$$R'(x) = \left\{ \begin{array}{ll} x & x \in [0, q] \\ q + \alpha(x - q) & x \in [q, 1] \end{array} \right.$$

Moreover, $c^* \in [C]$ such that $p_{c^*} = \pi$, is not included in $\Lambda'$. It is not hard to see that,

$$F(p', R', \Lambda') = \hat{F}(q) = \frac{\lambda}{\alpha(\lambda - 1) + 1} \cdot \frac{\alpha + (1 - \alpha)q}{\alpha + (1 - \alpha)Cq}.$$

Remark that $F(p', R', \Lambda')$ is decreasing on $q$. In particular, making $q \to q - \varepsilon$ increases the value of $F$. However, remark the value of $q$ defines the kink of the function $R'$. In addition, any decreasing on $q$ implies to decrease the value on the entries of $p$ within $\Lambda'$. Since $p$ is a probability distribution, the decrease of mass must be re-injected on all other entries whose values are below $\pi$, as we cannot modify the value of the highest-value entry of $p$. In conclusion, whenever solving

$$\max_{q \in [0,1]} \hat{F}(q),$$

we obtain the solution

$$q = \begin{cases} 0 \\ \frac{1-(C-\lambda)\pi}{\lambda} \end{cases}$$

where the second case comes from attaining the constraint of maximizing all entries $c \in [C] \setminus \{\Lambda \cup \{c^*\}\}$ up to $\pi$. Remark that when the previous optimization problem we do not consider anymore the space of functions $\mathcal{R}$, in particular allowing for $q = 0$ to be a possible solution. □

We are ready to prove Theorem 4.8.

*Proof of Theorem 4.8.* For $\lambda \in [C]$ and $\alpha \in [0,1]$, consider the function

$$\psi_\lambda(\alpha, q) = \frac{\lambda}{\alpha(\lambda - 1) + 1} \cdot \frac{\alpha + (1-\alpha)q}{\alpha + (1-\alpha)Cq}.$$

From Lemma E.6, it follows,

$$\max_{p \in \Delta_\pi^C} \max_{R \in \mathcal{R}} \max_{\Lambda \subseteq [C]} \frac{R(1)}{\sum_{c \in [C]} R(p_c)} \cdot \frac{\sum_{c \in \Lambda} R(p_c)}{R(\sum_{c \in \Lambda} p_c)} = \max_{\lambda \in [C]} \max_{\alpha \in [0,1]} \max_{q \in [0,1]} \psi_\lambda(\alpha, q). \tag{13}$$

We know that for $(\lambda, \alpha) \in [C] \times [0,1]$,

$$\underset{q \in [0,1]}{\operatorname{argmax}} \, \psi_\lambda(\alpha, q) = \begin{cases} 0 \\ \frac{1-(C-\lambda)\pi}{\lambda} \end{cases}$$

Suppose $1 - (C - \lambda)\pi \leq 0$. It follows,

$$\max_{p \in \Delta_\pi^C} \max_{R \in \mathcal{R}} \max_{\Lambda \subseteq [C]} \frac{R(1)}{\sum_{c \in [C]} R(p_c)} \cdot \frac{\sum_{c \in \Lambda} R(p_c)}{R(\sum_{c \in \Lambda} p_c)} \leq \max_{\lambda \in [C]} \max_{\alpha \in [0,1]} \frac{\lambda}{\alpha(\lambda - 1) + 1} = \max_{\lambda \in [C]} \lambda.$$

Suppose $1 - (C - \lambda)\pi \geq 0$, i.e., $q \in [0, 1/C]$. We study the first order conditions of $\psi_\lambda(\alpha, q)$ over $\alpha$. It follows,

$$\frac{d}{d\alpha} \psi_\lambda(\alpha, q) = -\frac{\lambda}{(\alpha\lambda - \alpha + 1)^2 (\alpha Cq - \alpha - Cq)^2} \left[ \alpha^2(C\lambda q^2 - C\lambda q - Cq^2 + Cq - \lambda q + \lambda + q - 1) \right.$$
$$\left. + \alpha(-2C\lambda q^2 + 2Cq^2 + 2\lambda q - 2q) + C\lambda q^2 - Cq^2 - Cq + q \right].$$

Imposing $\frac{d}{d\alpha} \psi_\lambda(\alpha, q) = 0$ we obtain the solutions,

$$\alpha_1 = \frac{2(-1+\lambda)q(-1+Cq) - \sqrt{4(-1+C)(-1+\lambda)q(-1+Cq)(-1+\lambda q)}}{2(q + Cq(-1 + (-1+\lambda)q))},$$

$$\alpha_2 = \frac{2(-1+\lambda)q(-1+Cq) + \sqrt{4(-1+C)(-1+\lambda)q(-1+Cq)(-1+\lambda q)}}{2(q + Cq(-1 + (-1+\lambda)q))}.$$

Since $q \leq 1/C$, it follows that $2(-1+\lambda)q(-1+Cq) \leq 0$ and, therefore, $\alpha_1 < 0$. The only possible solution being $\alpha_2$, we show that either

1. $\alpha_2 \in [0,1]$ and then the stated value of $\psi_\lambda(q)$ for $q \in \left(0, \frac{\lambda-1}{\lambda(C-1)}\right]$ comes from plugging $\alpha_2$ into Equation (13) or,

2. $\alpha_2 \geq 1$ and then Equation (13) is upper bounded by 1 for any $q \in \left(\frac{\lambda-1}{\lambda(C-1)}, \frac{1}{C}\right]$.

The first point is direct. For the second point, notice that $\alpha_2 \geq 1$ if and only if

$$\sqrt{4(C-1)(\lambda-1)q(Cq-1)(\lambda q-1)} \geq 2(\lambda-1)(p-1)(Cq-1) + 2(\lambda-1)q(1-Cq),$$

and, as $2(\lambda-1)(p-1)(Cq-1) + 2(\lambda-1)q(1-Cq) \geq 0$, this is equivalent to

$$4(C-1)(\lambda-1)q(Cq-1)(\lambda q-1) \geq (2(\lambda-1)(p-1)(Cq-1) + 2(\lambda-1)q(1-Cq))^2$$
$$\iff 4(\lambda-1)(1-q)(1-Cq)(\lambda((C-1)q-1)+1) \geq 0$$
$$\iff q \in \left(\frac{\lambda-1}{\lambda(C-1)}, \frac{1}{C}\right].$$

Since $\alpha_2 \geq 1$, the optimal value of $\alpha$ is 1, yielding

$$\max_{\alpha \in [0,1]} \max_{q \in [0,1]} \psi_\lambda(\alpha, q) = \max_{q \in \left(\frac{\lambda-1}{\lambda(C-1)}, \frac{1}{C}\right]} \psi_\lambda(1, q) = 1.$$

To conclude the proof, set $\psi_\lambda(q) := \max_{\alpha \in [0,1]} \psi_\lambda(\alpha, q)$. The relaxed upper bound

$$\max_{\lambda \in [C]} \psi_\lambda\left(\frac{1-(C-\lambda)\pi}{C}\right) \leq C - \frac{1}{\pi},$$

is obtained through symbolic computation in Mathematica (with the Reduce function). Indeed, it can be verified that the inequality system

$$\frac{\lambda(Cq-1)}{\lambda q-1} - \frac{\lambda\left(Cq(\lambda q+\lambda-2) - 2\sqrt{(C-1)(\lambda-1)q(Cq-1)(\lambda q-1)} - 2\lambda q+q+1\right)}{(C\lambda q-1)^2} \geq 0$$

for $q \in \left[0, \frac{\lambda-1}{C\lambda-\lambda}\right]$ and $\lambda \in [2, C-1]$,

is always feasible. It follows that $\lambda(Cq-1)/(\lambda q-1) \geq \psi_\lambda(q)$ for $0 \leq q \leq (\lambda-1)/(C\lambda-\lambda)$. In particular, for $q = (1-(C-\lambda)\pi)/\lambda$, it follows that $C - 1/\pi \geq \psi_\lambda(q)$ over $[1/(C-1), 1/(C-\lambda)]$. Similarly, since $C - 1/\pi$ is greater than $\lambda = \psi_\lambda(q)$ whenever $\pi \geq 1/(C-\lambda)$, we conclude $C - 1/\pi \geq \psi_\lambda(q)$ for any $\lambda \in [C]$ and $q \in [0,1]$. □

### E.11  A technical lemma

The following technical lemma gives a sufficient condition for a polymatroid to have a price of opportunity fairness equal to 1. In particular, several of the posterior results in the stochastic setting will use it.

**Lemma E.7.** *Let $M$ be a polymatroid. Given a permutation $\sigma \in \Sigma([C])$, consider the sequence $\mathrm{r}(\sigma) = (\mathrm{r}_c(\sigma))_{c \in [C]}$ such that,*

$$\text{for any } c \in [C], \mathrm{r}_c(\sigma) := \frac{\mathrm{r}(\sigma(1,...,c)) - \mathrm{r}(\sigma(1,...,c-1))}{\mathrm{r}(\sigma(c))},$$

*where, $\mathrm{r}(\sigma(1,...,c))$ corresponds to the size of a maximum size allocation in the submatroid obtained by the groups in the first $c$ entries of $\sigma([C])$. Whenever the sequences $\mathrm{r}(\sigma)$ for any $\sigma \in \Sigma([C])$, are all decreasing, it holds $\mathrm{PoF}(M) = 1$.*

*Proof.* Let $\Lambda^* = \mathrm{argmax}_{\Lambda \subseteq [C]} \frac{\sum_{c \in \Lambda} \mathrm{r}(c)}{\mathrm{r}(\Lambda)}$. We aim at proving that the monotonicity of the sequences $\{\mathrm{r}(\sigma), \sigma \in \Sigma([C])\}$ implies $\Lambda^* = [C]$, which yields $\mathrm{PoF}(M) = 1$. Without loss of generality, take $\sigma = I_C$ to be the identity permutation (the same argument works for any other permutation). Denote

$$\rho_t := \frac{\mathrm{r}([t])}{\sum_{\ell \in [t]} \mathrm{r}(\ell)},$$

the competition index of the submatroid obtained by the the first $t$ groups. Denoting $\mathrm{r}(0) = 0$, it follows,

$$\rho_{t+1} - \rho_t = \frac{\mathrm{r}([t+1])}{\sum_{\ell \in [t+1]} \mathrm{r}(\ell)} - \frac{\mathrm{r}([t])}{\sum_{\ell \in [t]} \mathrm{r}(\ell)}$$

$$= \frac{\sum_{\ell \in [t+1]} r(\ell) - r(\ell - 1)}{\sum_{\ell \in [t+1]} r(\ell)} - \frac{\sum_{\ell \in [t]} r(\ell) - r(\ell - 1)}{\sum_{\ell \in [t]} r(\ell)}$$

$$= \frac{\sum_{\ell \in [t]} [r(t+1) - r(t)] r(\ell) - r(t+1)[r(\ell) - r(\ell - 1)]}{\left(\sum_{\ell \in [t+1]} r(\ell)\right)\left(\sum_{\ell \in [t]} r(\ell)\right)}.$$

Since $r(\sigma)$ is decreasing, for any $s < t + 1$ it follows,

$$\frac{r(s) - r(s - 1)}{r(s)} \geq \frac{r(t+1) - r(t)}{r(t+1)}.$$

In particular, $r(t+1)[r(s) - r(s-1)] \geq [r(t+1) - r(t)]r(s)$ and therefore, $\rho_t \geq \rho_{t+1}$. It follows that the optimal solution corresponds to $\Lambda^* = [C]$. $\square$

### E.12 Proof of Propositions 4.10 and 4.11

**Proposition 4.10.** Let $\omega = \omega(n)$ be a function such that $\omega(n) \to \infty$. Whenever $q \leq 1/(\omega n)$ or $q \geq \omega/n$, for any $p \in \Delta^C$, $\mathrm{PoF}(G_{n,q}(p))$ converges to 1 with high probability as $n$ grows.

*Proof.* We show this proposition by leveraging results from random graph theory. Suppose $q \leq 1/(\omega n)$. By Theorem 2.1 [37], $G_{n,q}$ is a forest w.h.p.. It follows that, independent of the label realization, the maximal allocation contain all edges in the graph. Hence $\sum_{c \in [C]} r(c) = r([C])$, which implies the matroid has PoF equal to 1 as the independence index is equal to 1.

Suppose $q \geq \omega/n$. For any $c \in [C]$ the subgraph induced by considering only the subset $E_c$ over $G_{n,q}$ is distributed according to $G_{n,p_c q}$. Since $p_c q \geq p_c \omega/n$, with $p_c \omega \to \infty$ arbitrarily slow, Theorem 2.14 [37] states that w.h.p. $G_{n,p_c q}$ has a giant component of size $(1 - \frac{x}{p_c \omega})n$, for a fixed $x \in [0,1]$. In particular, $r(c) = (1 - \frac{x}{p_c \omega})n$ as connected components contain spanning trees. Intersecting the events over all $c \in [C]$ we obtain, w.h.p. as $n$ goes to infinity,

$$\rho(G_{n,q}(p)) = \frac{r([C])}{\sum_{c \in [C]} r(c)} = \frac{n}{\sum_{c \in [C]} n} + o(1) = \frac{1}{C} + o(1),$$

which from Proposition 4.6 shows that PoF is also equal to 1 w.h.p.. $\square$

**Proposition 4.11.** Let $\omega = \omega(n)$ be a function such that $\omega(n) \to \infty$ arbitrarily slow as $n \to \infty$. Whenever $q \leq 1/(\omega n^{3/2})$ or $q \geq \omega \log(n)/n$, for any $p \in \Delta^C$, $\mathrm{PoF}(B_{n,\beta,q}(p))$ converges to 1 with high probability as $n$ grows.

*Proof.* Suppose $q \geq \omega \log(n)/n$. Let $\Lambda \subseteq [C]$, we have that $\sum_{c \in [C]} |E_c|$ is a sum of independent bernouli random variables (the $E_c$ are disjoints), hence it has an expected value of $n \sum_{c \in \Lambda} p_c$ and Hoeffding's concentration inequality show that

$$\mathbb{P}\left(\left|\sum_{c \in \Lambda} |E_c| - n \sum_{c \in \Lambda} p_c\right| > \sqrt{n \log(n)}\right) \leq 2 \exp\left(-2 \frac{n \log(n)}{n}\right) = \frac{2}{n^2} \xrightarrow[n \to \infty]{} 0.$$

Since $q \geq \omega \log(n)/n$, Theorem 6.1 [37] states that w.h.p. for any $\Lambda \subseteq [C]$, the subgraph considering only the vertices in $\Lambda$ on the left-hand side has a matching of size $\min\{\beta n, \sum_{c \in \Lambda} p_c n\}$, therefore, $r(\Lambda) = \min\{\beta n, \sum_{c \in \Lambda} p_c n\}$. We will conclude by applying Lemma E.7. As usual, w.l.o.g. consider $\sigma = I_L$. For any $c \in [C - 1]$, it follows,

$$r_{c+1}(\sigma) = \frac{\min\{\beta n, \sum_{c' \in [c+1]} p_{c'} n\} - \min\{\beta n, \sum_{c' \in [c]} p_{c'} n\}}{\min\{\beta n, p_c n\}}.$$

In particular, as $\sum_{c' \in [c]} p_{c'}$ is increasing in $c$, the sequence $r_{c+1}(\sigma)$ initially consists on only 1 (given by all times that $r_{c+1}(\sigma) \leq \beta$), eventually some value between 0 and 1 (given by the first time that $r_{c+1}(\sigma) \geq \beta \geq r_c(\sigma)$), and finally a sequence of only zeros (given by all times when $r_{c+1}(\sigma) > \beta$). In particular, the sequence is decreasing, concluding the proof. $\square$

**Remark 2.** In both graphic and transversal random matroids, taking the same $q \in [0, 1]$ for all colors is done without loss of generality. Indeed, a coupling argument based on stochastic dominance allows us to consider edge probabilities $q_c$ per color $c$ and to obtain the same results.

# F  Price of Fairness under Other Fairness Notions

## F.1  Weighted Fairness

The main fairness definition that we have used is opportunity fairness. We now discuss how this specific fairness notion relates with other fairness concepts, in particular with maximin fairness and proportionality in [11] and equitability in [15]. We can think more generally about group fairness in terms of what amount of social welfare protected group of agents are entitled to. Should each group be entitled to the same amount as others, or proportionally to their size? We introduce weighted fairness, where the weights correspond to group entitlement:

**Definition F.1.** Let $(w_c)_{c \in [C]} \in \mathbb{R}_+^C$ be a fixed weights vector. An allocation $x \in \mathbb{R}_+^C$ is $w$-fair if for any $i, j \in [C]$, $x_i/w_i = x_j/w_j$.

As an example, Figure 20 illustrates for a 2-colored matroid three fairness notions mentioned in the paper that are now framed as specific instances of weighted fairness.

1. **Equitability**
   $w_c = 1$ for $c \in [C]$,
2. **Proportional fairness**
   $w_c = |E_c|$ for $c \in [C]$,
3. **Opportunity fairness**
   $w_c = \mathrm{r}(c)$ for $c \in [C]$.

Figure 20: Weighted Fairness for matroid with two groups

Compared to other weighted fairness notions, opportunity fairness remains bounded because the weights depend on the structure of the polymatroid $M$, while the weights of proportional fairness and equitability are independent of $M$ and arbitrarily bad examples can easily be constructed.

**Proposition F.2.** *The price of proportional fairness and the price of equitability are unbounded in the worst-case, even when allowing for fractional allocations.*

*Proof.* Take a ground set with $n$ agents of color $1$ and $n$ agents of color $2$, and consider the feasible allocations where at most $n/2$ resources can be allocated to individuals of color $2$ but only one resource may be allocated to individuals of color $1$. This is a partition matroid, where the partition corresponds exactly to the color partition. Now, the optimal proportionally fair allocation, as well as the optimal equitable allocation, is $x_1 = 1$ and $x_2 = 1$, for a total of 2 resources allocated. Because the optimal allocation is of size $n + 1$, the price of fairness is $(n+1)/2$, which goes to $\infty$ as $n \to \infty$. The price of fairness in both cases is unbounded.  $\square$

Another common concept of fairness to divide resources, used in transferable utility cooperative game theory, is that of Shapley value [58]: it is the unique utility transfer that satisfies axioms of symmetry, additivity, nullity and efficiency. It can be shown that for $\Sigma([C])$ the set of permutations over $[C]$ the Shapley value of group $c$ is

$$\varphi_c := \frac{1}{C!} \sum_{\sigma \in \Sigma([C])} \mathrm{r}(\{i \in [C] \mid \sigma(i) < \sigma(c)\} \cup \{c\}) - \mathrm{r}(\{i \in [C] \mid \sigma(i) < \sigma(c)\}),$$

that is to say $\varphi_c$ is the expected marginal contribution of group $c$ when groups are prioritized according to $\sigma$ a uniformly drawn random permutation. When $w_c = \varphi_c$, we say that an allocation is Shapley fair.

The allocation problem we study can be seen as a type of non transferrable utility game, and as such there is no reason in general for the allocation $(\varphi_1, \dots, \varphi_C)$ to be realizable. Nonetheless, from the polymatroid characterization of $M$ do have this property:

**Proposition F.3.** *The allocation $(\varphi_1, \dots, \varphi_C)$ is always feasible.*

*Proof.* For a given permutation $\sigma$, the marginal contribution allocation $x^\sigma$ wher $x_c^\sigma = \mathrm{r}(\{i \in [C] \mid \sigma(i) < \sigma(c)\} \cup \{c\}) - \mathrm{r}(\{i \in [C] \mid \sigma(i) < \sigma(c)\})$ is always feasible. Hence, the Shapley allocation $(\varphi_1, \ldots, \varphi_C)$ the barycenter of all the $x^\sigma$, which belong to the Pareto front by definition. Moreover, by the polymatroid characterization, the Pareto front is convex [43]. Hence the barycenter, being a convex combination, is also feasible. We note that the $x^\sigma$ are the extreme points of the Pareto front. $\qquad\square$

From the efficiency of the Shapley allocation, it is immediate that the Shapley Price of Fairness is always 1 for colored matroids.

In the semi-random model of Theorem 4.8, we can easily show the following property:

**Proposition F.4.** *For any distribution $p \in \Delta^C$ of the agents colors, in the large market setting with $\liminf \mathrm{r}_n([C]) = \Omega(n)$, we have that the price of proportional fairness converges to* 1 *with high probability.*

*Proof.* Let $S_n$ be any maximal allocation, taken independently of the random coloring. We have that $|S_n| = \Omega(n)$ and $|S_n \cap E_c|$ concentrates towards $p_c|S_n|$ by Hoeffding's inequality. We also have that $|E_c|$ concentrates around $p_c n$. Hence $|S_n \cap E_c|/|E_c|$ concentrates around $|S_n|/n$, which is independent of the colors, and therefore is a proportionally fair allocation. Moreover $S_n$ is maximal by definition, so the price of fairness is equal to 1. $\qquad\square$

This shows that the price of proportional fairness goes from $\mathrm{PoF} = +\infty$ in the adversarial setting to $\mathrm{PoF} = 1$ in the semi-random setting.

Finally let us mention another fairness definition.

## F.2 Leximin Fairness

Requiring that a fair allocation satisfies exactly $x_i/w_i = x_j/w_j$ can be considered wasteful, as it is possible to improve the total social welfare without making any group worst off. Maximin fairness [11], also called egalitarian rule or Rawlsian fairness, corresponds to ensuring that the worst off group has the best allocation possible. In other words, an allocation is maximin fair if it maximizes $\min_{x \in M} x_c$, or with entitlement $w$, maximizes $\min_{x \in M} x_c/w_c$. Most of the time there are multiple maximin fair feasible allocations, and thus one may seek to maximize the second minimum, and so forth. This is called the leximin rule, and has also been studied in the social choice literature [25, 27].

For a vector $x = (x_1, \ldots, x_C)$, we denote the ordered coordinates by $x_{(1)} \geq x_{(2)} \geq \cdots \geq x_{(C)}$. We say that a vector $x = (x_1, \ldots, x_C)$ is leximin larger than $y = (y_1, \ldots, y_C)$ if $x_{(C)} \geq y_{(C)}$, or $x_{(C)} = y_{(C)}$ and $x_{C-1} \geq y_{(C-1)}$, or $x_{(C)} = y_{(C)}$ and $x_{C-1} = y_{(C-1)}$ and $x_{C-2} = y_{(C-2)}$ and so on. The leximin order is a total preorder. Leveraging the more general notion of weighted fairness, we have the following definition:

**Definition F.5.** For a weight vector $w \in \mathbb{R}_+^C$, an allocation $(x_1, \ldots, x_c)$ is said to be $w$-lexmaxmin fair if $\left(\frac{x_1}{w_1}, \ldots, \frac{x_C}{w_C}\right)$ is maximal according to the leximin order for $x \in M$.

Clearly, the $w$-lexmaxmin fair allocation is $w$-maxmin fair. It is also Pareto efficient, and therefore by the polymatroid characterization Proposition 3.2 achieves maximal social welfare: the price of $w$-lexmaxmin fairness is always 1 in $C$-colored matroids.

