# OpenReview forum: "The Price of Opportunity Fairness in Matroid Allocation Problems"
_NeurIPS.cc/2025/Conference — NeurIPS 2025 poster_

### Official Review · Reviewer_JNtD · 2025-06-13

**Clarity:** 3
**Significance:** 3
**Originality:** 3
**Rating:** 5
**Confidence:** 3

**Summary:**

The paper focuses on a fair allocation problem subject to matroid constraints where (1)there is a set E of agents each of which either gets allocated an item or does not get; (2) a set I of agent subsets is specified as a set of “feasible allocations” and such agent set system (E,I) forms a matroid; (3) a feasible allocation is optimal if it contains the maximum number of agents among all feasible allocations and the number of agents in an allocation is defined as social welfare; (4)the agent set are assumed to already be partitioned into several groups according to a sensitive attribute and an allocation x is considered to be “opportunity fair” (newly proposed by the paper) if the ratio: “# of agents from group c in x / maximum number of agents in a feasible allocation containing only agents from group c” are all the same for each group c. The paper studies the price of opportunity fairness defined as the ratio between the cardinality of an optimal allocation and the cardinality of a fair allocation.
In four contexts the paper gives bounds of PoF. First is normal worst case bound which depends on a specific “polarized” matroid structure; secondly the paper uses a parameter measuring the degree of balancedness of matroid to bound the PoF; thirdly the paper studies the semi-random scenario where each element in the matroid is assigned a color under some probability distribution; lastly the paper studies the complete random scenario in the sense that both matroid and color are constructed randomly. This is achieved based on a graph matroid of a random graph.

**Questions:**

Weakness on Writing
The writing in both main body and the proofs is a bit too technical and could be better organized to improve the readability, making readers more efficiently get the exact settings of the problem and the logic of the proofs. The writing is a bit abstract and not easy to follow in some places with many jargons. Adding more intuitive explanations can be helpful.
- For example, the paper names the problem they study as “matroid allocation problem” and uses that phrase in the title. This is somewhat vague and potentially misleading. In the literature such problems are usually referred to as “selection problems” rather than “allocation problems”. Usually I expect that allocation problems involve more items than agents, where each agent may receive multiple items and preferences are involved. But in this paper’s setting, each agent can only get at most one item, and the goal is to select a bunch of agents who can get something simultaneously. Additionally, the fairness notion the paper introduces is a bit vague in the introduction. These factors left me very confused about the problem after reading the introduction. No concrete example is provided to aid understanding.
- The paper is proof-heavy, but in many instances, it jumps directly into details without offering an overview or explaining the intent, making the proofs difficult to follow.
- The model section (Section 2) is loosely organized. For example, the not-short comparison of fairness definitions inserted into the model section is somewhat distracting, though it does offer insights. Since the introduction did not give me a clear view of the problem, I hoped the model section would, but it took many words and effort to piece together the full setting. I had to summarize four pages myself to grasp the core idea.

Minor errors in spelling and format:
(1) Between line 58 and 59, should add “of” after # on numerator.
(2) Line 1012: “sub-addivitiy” should be “sub-additivity”.
(3) Line 168: min_{c\in[C]} r_c, r_c should be r(c) since you didn’t define r_c before.
(4) Line 188: the range of submodular function should be $\mathbb{R}_+$ instead of $\mathbb{R}_+^C$.
(5) Line 978: the second r([C]) should be r(c).
(6) In appendix E.10 proof of Theorem 4.8 and Lemma E.2, the proof environments are missed. In Line 1136 “consists on” should be “consists of”.

**Ethical Concerns:**

["NO or VERY MINOR ethics concerns only"]

**Final Justification:**

The authors have addressed my major concern but I am not very sure about the quality of the revision the author can provide. I maintain my score.

**Limitations:**

yes

**Paper Formatting Concerns:**

No.

**Quality:**

3

**Strengths And Weaknesses:**

Strength
- The four contexts under which PoF is measured are rich and representative, offering a comprehensive picture of its bounds. The logic flow between them is intuitive and makes good sense.
- The proposed opportunity fairness notion generalizes and complements proportionality fairness and is suitable for matroid setting, well considering and eliminating the effect of unrelated agents.
- As far as I know, there is little literature studying PoF for problems of this kind. This paper compensates for that gap.

Weakness
- As the paper itself points out, it uses fractional fair allocation to derive bounds on PoF, which introduces negligible inaccuracy when the smallest group is large but may not be accurate otherwise. But this weakness is acceptable and understandable to me.

---

> ### Author Rebuttal · Authors · 2025-07-30
>
> We thank the reviewer for their thoughtful review and appreciation of our contributions, particularly regarding the opportunity fairness notion and the beyond-worst-case analysis of the PoF. We address below the reviewer’s concerns regarding the terminology, clarity, and presentation.
>
> ### **1. “Matroid Allocation” vs. “Selection” Terminology**
>
> We agree with the reviewer’s point that, in combinatorial optimization, the problems we study are often referred to as selection problems. Our use of the term `matroid allocation’ was motivated by a common framing of these types of problems from the market design (in particular bipartite matching) and fair resource allocation literature: although the decision is to select a feasible subset under matroid constraints, our objective is to allocate limited resources fairly across groups (see e.g., Group-level Fairness Maximization in Online Bipartite Matching, Ma, Xu, and Xu where they also use the term allocation). Additionally, the setting we study can also be seen as allocating resources to the groups directly, which is also basically the main way we tackle this problem from a technical aspect, by forgetting agents and only focusing on the group allocation.
>
> That said, we acknowledge the potential confusion. We will revise the introduction to: Clearly state that each agent can receive at most one item, clarify the connection with selection problems, and mention the aspect of allocating resources to groups directly.
>
>
> ### **2. Clarity of Fairness Definition**
>
> We agree that the current model section lacks a concrete motivating example, and we will include a  worked out bipartite matching example with $C=3$ added at the end of subsection 2.2. We will describe the set of opportunity fair allocations, compute the optimal unfair allocation, the optimal fair allocation, and the PoF. We will also move move the current subsection comparing fairness notions (2.3) to the end of Section 2, so as not to interrupt the model setup,
>
> ### **3. Proof heavy paper**
>
> We recognize that many proofs are dense and technical. We will include for key proofs (especially Proposition 2.5, Proposition 3.2, and Theorem 4.1) additional insights, high-level overview, and roadmaps, before diving into the technical details.
>
> ### **4. Typos**
>
> Thank you for catching these typos, they will all be corrected in the final version.
>
>
> We acknowledge the reviewer’s feedback that our paper was challenging to follow in some places. Using the additional page allowed for the final version, we will implement all the above changes to improve readability, clarify terminology, and proofs overview. We are grateful for the reviewer’s input and would be happy to discuss any additional points during the rebuttal period.

---

> > ### Comment · Reviewer_JNtD · 2025-08-05
> >
> > I would like to thank the authors for the reply. About the term of “matroid allocation” which makes me confused before, your reply that the problem can be seen as allocating resources across groups makes sense to me and in that sense the problem is ok to me to be called an allocation problem. And you said you will mention this clarification in revised version so I have no further problem.

---

### Official Review · Reviewer_SYAG · 2025-06-30

**Clarity:** 2
**Significance:** 3
**Originality:** 3
**Rating:** 4
**Confidence:** 3

**Summary:**

This paper studies the matroid allocation problem with the fairness constraint - opportunity fairness, and focuses on the price of fairness (PoF) to capture the loss when the opportunity fairness is added. First, they build the link between fractional allocation and integral allocation when considering the opportunity fairness in the large market, which gives us sufficient reasons to consider fractional allocation. Then, they give the characterization of the PoF for a polymatroid, which is the foundation of the following analysis. Next, they bound the PoF in adversarial, parametric, semi-random, and random graph settings and present a series of results.

**Questions:**

Q1: In the abstract, the authors said “matroid constraints include classical problems such as bipartite matching”, but in my personal opinion, bipartite graph matching is not a matroid because it does not meet the augmentation property.

Q2: This paper proposed the concept of opportunity fair. Can you give an example to illustrate this fairness? Is it related to the “($\alpha$, $\beta$) – balance” mentioned in reference [17]? I haven't seen any papers related to this concept in the related work.

Q3: Can the “C-colored matroid” be understood as the intersection of two matroids?

**Ethical Concerns:**

["NO or VERY MINOR ethics concerns only"]

**Final Justification:**

The authors have addressed my concerns. Although there are a lot of negative results, the contribution is still solid. I will keep my score.

**Quality:**

3

**Strengths And Weaknesses:**

Strength:

They address matroid allocation problems under a novel group-fairness notion—opportunity fairness—and prove tight bounds for the price of fairness. They give insights into which aspects of the problem’s structure affect the trade-off between opportunity fairness and social welfare. It is meaningful. There are many positive results in multiple settings from adversarial to fully random.


Weakness:

The model description is not clear enough. For example, the expression for social welfare considered in the article is not indicated in the model introduction. Lack of explanation of the definition when proposing the new concept of fairness.

---

> ### Author Rebuttal · Authors · 2025-07-30
>
> We thank the reviewer for their careful review and helpful comments. We answer below the reviewer’s questions.
>
> - **(Q1)** Whether a bipartite matching problem corresponds to a matroid depends on the exact formulation. If the ground set is taken to be the set of edges, and feasible sets are matchings, the problem corresponds to the intersection of two partition matroids, and is not itself a matroid. However, if the ground set is instead the set of left-side nodes (e.g., agents), and feasible sets are chosen as the sets of matched left-side nodes for each feasible matching, then the constraint defines a transversal matroid. Since our paper studies fairness at the level of agents (not edges), the underlying feasibility constraint is indeed a matroid. The formal formulation can be checked in Appendix B of the article.
>
> - **(Q2)** The main idea of this fairness notion is to give entitlements to different groups depending on their underlying quality. As an example, we can consider refugees resettlement, with $n$ men and $m$ women, which can be matched in a bipartite graph to different cities. If only the men were present, $r_1$ men could be relocated, while if only women were present, $r_2$ women could be relocated. The maximal matching is of size $r_{1,2} \leq r_1 +r_2$. Under opportunity fairness, each group receives a share of the maximal matching size that they could achieve in isolation, which are respectively $r_1$ and $r_2$ for men and women. For this instance, the optimal opportunity fair matching, which equalizes $x_1/r_1=x_2/r_2$ will match $r_1 r_{1,2}/(r_1+r_2)$ and $r_2 r_{1,2}/(r_1+r_2)$ women. Whereas the proportionally fair matching, which equalizes $x_1/n=x_2/m$, would match $\min(r_1,r_2 n/m, n/(n+m) r_{1,2})$ men and $\min(r_2,r_1 m/n, m/(n+m) r_{1,2})$ women. To an extent, proportional fairness depends heavily on the number of men and women, whether they can actually be matched or not. The notion of $\alpha,\beta$ fairness of $[17]$ is more closely related to equitability, as the $\alpha$ and $\beta$ are independent of $c$, and $x_c$ is normalized by $\sum_c x_c$ in their paper. In the setting we consider, $x_c$ is normalized by $r(c)$.
>
> - **(Q3)** A $C$-colored matroid is not to be understood as an intersection of two matroids, but rather as a polymatroid, as shown in Proposition 3.2.
>
> We will revise the model section to explicitly define the social welfare objective and clarify the definition of opportunity fairness.
>
> Please feel free to ask any additional questions during the rebuttal period.

---

> > ### Comment · Reviewer_SYAG · 2025-08-06
> >
> > Thanks for your response! It is much clearer to me. I will maintain the score.

---

### Official Review · Reviewer_JYFo · 2025-07-02

**Clarity:** 4
**Significance:** 2
**Originality:** 3
**Rating:** 4
**Confidence:** 3

**Summary:**

This paper studies a new notion of fairness, which they term "opportunity fairness", for the problem of selecting a maximal independent set in a matroid.  In this setup, the elements $E$ of the universe are partitioned into $C$ groups.   Given an independent set $I$, they say that this set is "opportunity fair"  if for every two groups $c_1, c_2$, it is true that $|I \cap c_1| / r(c_1) = |I \cap c_2| / r(c_2)$.  In other words, if a group $c$ has large rank, then it should accordingly be highly represented in the chosen independent set.  On the other hand, if $c$ has low rank, then it should not appear as much in the independent set.  By assuming that every group has large rank, this paper extends this definition to the fractional setting with only negligible loss, and then mainly focuses on the fractional setting.

The obvious comparison for this notion of fairness is to proportional fairness, which looks at the ratios $|I \cap c_1| / |c_1|$ rather than $|I \cap c_1 / r(c_1)|$.  As the authors point out, proportional fairness has the bad property that if we simply add lots of loops to the matroid (elements e where $\{e\}$ is not an independent set) in group $c$ then we can force $|c|$ to be large and hence force $|I \cap c| / |c|$ to be small, which then would require for fairness that all other groups also get a small number of elements into $I$.  While this can of course be easily avoided by preprocessing to remove loops, this issue is more general -- it is trivial to construct matroids where a group has large size but low rank, and this would then require all other groups to have small representation in $I$ in order to achieve proportional fairness.  This new definition gets around this problem by normalizing by the rank.

This studies the "price of fairness" for opportunity fairness, and proves a number of results:
- In the worst case the PoF is extremely large: $C-1$.
- If all groups have about the same rank then the PoF is more like $C/4$
- More refined bounds on the PoF in terms of the "independence index" of the matroid
- In a semi-random setting where the adversary picks the matroid but the groups are random (there is a distribution over groups and each element draws from this distribution to determine its group), there is a very complex bound on the PoF, which (as far as I can tell) gives nontrivial bounds whenever the max probability from this distribution is reasonably small.
- If we also make the matroid random, then for ranom graphic and transversal matroids (defined by Erdos-Renyi random graphs or bipartite versions thereof) the authors prove even stronger bounds.

**Questions:**

- What are other examples where this setup makes sense?  Or is it really only matching markets?
- Similarly, can you provide examples or justification for the semi-random model?
- In the semi-random model, why do you have an infinite series of matroids?

**Ethical Concerns:**

["NO or VERY MINOR ethics concerns only"]

**Final Justification:**

I basically like this paper, but I was not particularly convinced by the reviewer's response about either the semi-random model or the examples of matroid for justifying the problem.

**Limitations:**

yes

**Quality:**

2

**Strengths And Weaknesses:**

Strengths:
- I like this notion of fairness -- it feels very natural, and well-motivated at least for some settings (e.g., the job allocation problem that they mention is their motivating example).  And once there's a nice notion of fairness, studying the PoF is a natural question.
- The mathematical depth and rigor is above the bar.
- The paper is quite well-written, with very nice and helpful figures.

Weaknesses:
- I don't quite see the motivation for the general version of this problem that they've stated (As opposed to for specific matching problems like job allocation).  I wrote the whole summary without using the word "allocation", because as far as I can tell nothing in the paper is actually about allocations.  They just assume that the feasible allocations form a matroid, and from then on we don't have to think about allocations at all.  This is a reasonable thing to do, and in similar situations has led to lots of super interesting work (e.g., on matroid secretary problems).  But in the context of fairness, I wasn't actually convinced by any of their other examples.  As far as I can tell, their examples are the following (see top of page 2):
	- Matching markets, which correspond basically to transversal matroids.  This motivation makes sense to me, and as the authors point out, there has been previous work on fairness in matching markets.
	- Communication networks between cities, where we want to ensure fairness in order to have competition between providers.  They say that this corresponds to graphic matroids for their problem, but I don't really see this.  Presumably  with this motivation we want to select not an independent set, but rather a full-rank subset (including a base) since we don't want a non-connected forest!  So the problem doesn't seem to fit.  And I'm not sure how fairness would correspond to competition -- that seems like quite a leap economically.
	- "the selection of members for a constitutional commission that must satisfy parity requirements such as gender and ethnic representation", which they say corresponds to a uniform matroid.  But in a uniform matroid the rank of each group is the minimum of the rank of the matroid and the size of the group, so under their large-market assumption the ranks would all be the rank of the matroid itself and so this just boils down to the question of selecting an equal number from each group (which is trivial).
- Similarly, as the authors state in their limitations section, they cannot handle the weighted scenario.  This feels to me like an extremely important limitation -- in the vast majority of economic situations that I'm aware of (where one would talk about allocations like this) there are weights involved (see, for example, the matroid secretary literature mentioned above).
- The semi-random model feels very strange to me.  Why would the assignment to groups be IID?  The authors don't provide any example or justifications (as far as I can tell).
  - The technical setting for the semi-random model is also very strange to me.  Consider equation (2), where they require that a limit exists as the size of the matroid goes to infinity.  But this means that (as they say) they have an infinite series of matroids.  Given an arbitrary infinite set of matroids, why would one expect the normalized rank to be at all stable?  Maybe the matroid of size 1000 is a low-rank uniform matroid, the matroid of size 1001 is a high-rank transversal matroid, the matroid of size 1002 is a random paving matroid, the matroid of size 1003 is a gammoid, etc.  I don't see why one would expect a limit for the ranks to exist, when the matroids are chosen adversarially!  It seems to me like the much more natural semi-random setting would be for there to be a single adversarial matroid and then a random coloring of it.  Why is there an infinite sequence of adversarially chosen matroids that has to have a limit?
- The random graphical and transversal matroids are fun mathematical exercises, but again, I'm not sure why they're important or what the example is where one would actually expect to see such a random graph.

---

> ### Author Rebuttal · Authors · 2025-07-30
>
> We thank the reviewer for their careful reading of the paper and their appreciation of the fairness measure we introduce. Below, we address the two main concerns raised by the reviewer:
>
> ### **1. Motivation for Matroids**
>
> We expand here on both the technical and modeling motivations for using matroids in our framework.
>
> First, as stated in line 38, our primary application motivation indeed comes from bipartite matching settings (e.g., job markets or refugee resettlement). However, our core results rely on matroid and polymatroid structure, which provide well understood structural properties that significantly simplify the analysis, and pinpoints the important structural properties. For example, an earlier proof of Corollary 3.3 relied on augmentation paths and was much more complex. Working with matroids allows us to leverage structural tools like submodular rank functions and base polytopes, which is at the heart of the analysis.
>
> Additionally, working with general matroids gives better insights on which instances are hard or easy. For instance, any lower bound on the PoF for a subclass (e.g., graphic matroids) automatically implies a lower bound for broader classes. As noted on lines 992–998, uniform matroids always have PoF = 1, whereas graphic and partition matroids, which are the two smallest natural families of matroids, have a worst-case PoF of $C - 1$.
>
> To further motivate the use of matroid in fair resource allocation, we now provide three additional examples beyond graphic and bipartite settings:
>
> - **Partition Matroids:**  Consider the set of all professors in a university, divided by department. The university can allocate research funding to the professors such that the aggregated funding of each department is capped. Additionally, the university may try to be fair by equally treating junior and senior professors, or male and female professors. This setting can be modeled as a partition matroid with fairness constraints.
>
> - **Laminar Matroids:** Consider a university organized into departments, each of which is further divided into teams. Suppose that each team has a capacity, corresponding to the number of people that can be accommodated in its offices. This situation can be modeled as a laminar matroid, where the sets in the laminar family correspond to the teams and departments, and the capacity function reflects the available office space. Our framework can be used to model scenarios in which the university aims to open a fixed number of faculty positions in a given year and seeks to allocate them across teams and departments in a fair manner.
>
> - **Linear Matroids:** Suppose a manager wants to hire a team with complementary skills from a set of candidates. This situation can be modeled as a linear matroid, and our framework can be used to study fairness in the selection process.
>
> For Linear Matroids, similarly to graphic matroid as correctly pointed out by the reviewer, it makes more sense to consider a full-rank subset. This corresponds to the dual matroid, yielding co-graphic matroid, where feasible sets are sets of edges that can be deleted while leaving the graph connected, and linear matroid, being closed under duality.
>
> We will include these examples into the final version of the paper by using the additional page, to better justify the modeling choice and expand the applicability of our results beyond traditional bipartite matching settings.
>
> ### **2. Semi-Random Model Interpretation**
>
> One of our key motivations in introducing the semi-random model is to understand what structural assumptions lead to high or low unfairness. In this model, the groups are random, while the matroid is adversarially chosen. This corresponds to a scenario in which agents have equal access to opportunities ex ante, or in other words, the underlying quality of an agent is independent of its group. Any observed inequality arises from feasibility constraints and group sizes, not intrinsic group differences.
>
> To provide further intuition, we can view the semi-random model as a stream of incoming agents, arriving one by one, where each agent has different opportunities based on time of arrival (e.g. advertisers currently running ad-campaigns), and the group assignment is uncorrelated with those opportunities.
>
> Regarding the reviewer’s suggestion of analyzing a single adversarial large matroid with random groups, this is in fact closely aligned with our approach. Matroid sequences allow us to study asymptotic limits directly, rather than having to deal with $O(\sqrt{n})$ error terms stemming from the concentration bounds. Importantly:
>
> - **Convergence:** Since the normalized rank function of any matroid is bounded in $[0,1]$, any sequence of matroids has a converging subsequence. Thus, we can focus on the limiting behavior, up to a converging sub-sequence, as suggested in lines 294–296.
>
> - **Different matroid types:** Although the matroids in the sequence may differ structurally, Proposition 4.7 essentially shows that the PoF asymptotically depends only on the rank function, due to the randomness in group assignments. This abstracts away from the matroid’s precise structure and focuses on the fairness-relevant quantities.
>
> We thank the reviewer for giving us the opportunity to clarify those points, and we will add these remarks into Section 4.3 in the final version.
>
> We are happy to continue this discussion with the reviewer during the rebuttal period, and answer any additional questions.

---

> > ### Comment · Reviewer_JYFo · 2025-08-04
> >
> > Thank you for the response.  I basically liked this paper, and still like it.  But I was not particularly convinced by the response on the semi-random model.  "Up to a converging subsequence" is doing a lot of work in both the original paper and the response, and I still think that it's far more natural and informative to handle a single adversarial matroid and then maybe talk about random assignment (like is done in the matroid secretary problem).  So I'm going to leave my score where it is.

---

### Official Review · Reviewer_vFMq · 2025-07-02

**Clarity:** 3
**Significance:** 3
**Originality:** 3
**Rating:** 4
**Confidence:** 3

**Summary:**

The paper studies matroid allocation problems under opportunity fairness constraints, a novel group-fairness notion, which states that each group gets the same share proportional to the max it could achieve in isolation. The quality of the allocation under such constraints is measured via the well-established Price of Fairness (PoF), which quantifies the social welfare loss due to the fairness constraints. The authors work with fractional allocations, since they argue that we can always get a random allocation that produces an integral allocation at each realization, with very small loss in the PoF and violation of the fairness constraints under a large market assumption.

The authors prove tight PoF bounds in a variety of regimes:
i) For adversarial matroid selection and group partition, they prove a tight bound that depends linealry on the number of groups, and is independent of the other parameters of the problem. For obtaining potentially better bounds, they work with specific parameters that relate to the shape of the underlying polymatroid and provide parametric bounds; one bound has to do with the ranks of the groups, and the other with the level of competition among groups for the same resources.
ii) Moving beyond worst-case, the authors first consider the regime in which there is an adversarial matroid choice with a random group partition of the agents. Here, the main message is that when there is no “dominant” group, there is no social welfare loss. When that’s not the case, the worst-case PoF bound is still linear in the number of groups.
iii) Finally, for random groups, and for random Erdos-Renyi graphs for graphic and traversal matroids, under some conditions on the randomness of the graph, they recover the result on PoF of 1 while being opportunity fair, without now the assumption that there cannot be any dominant group.

The results are based on the important observation that the set of feasible per-group allocations can be represented as a polymatroid, which gives a new characterization for the price of opportunity, and subsequently enables/simplifies the corresponding proofs.

They also show lower and upper bounds for $\gamma$-opportunity fairness, so solving for perfect fairness can automatically give bounds for approximate fairness as well.

**Questions:**

See the main questions under weaknesses, and some smaller ones for clarifications below:

Do you have to assume in proposition 4.6 that all groups have equal ranks? Can the results extend? Does this imply any limitation in the result and the message the paper tries to convey with this parametric bound?

Could you provide some intuition on Proposition 2.5, how we ended up with this upper bound and how the gap in the bounds changes as we vary $C$ and $\gamma$?

In the semi-random setting without the no-dominant-group assumption, this $C-2$ bound is worse than the adversarial setting because the analysis here is non-tight, or what am I missing?

A small suggestion: For subsection 4.4, either allocate a bit more space to explain the results and the assumptions, or move all technical statements to the appendix, and provide only intuition and key takeaways in the main body.

**Ethical Concerns:**

["NO or VERY MINOR ethics concerns only"]

**Final Justification:**

My questions during the rebuttal were resolved. As I mentioned, overall, I'm positive about the paper. I still have some concerns about its practical implementation (e.g., regarding the time for solving very large instances, the ex-post guarantees, the weighted scenario not being handled, as pointed also by reviewer JYFo), and thus, I have decided to keep my original score.

**Limitations:**

yes

**Paper Formatting Concerns:**

No concerns

**Quality:**

3

**Strengths And Weaknesses:**

Strengths:

I like the generality of the results, i.e., that for the adversarial (resp. the semi-random setting), the results apply for any type of matroid and any partition into groups (resp. distribution that randomly divides agents into groups). I also like the fact that most bounds are tight, and that the authors put the effort to construct tight examples for the bounds, as e.g., in Theorem 4.1 and Proposition 4.6.

The paper is carefully written, with clear statements, clean notation and precise proofs, some of which have technical depth. For instance, I found interesting the steps of reducing the infinite-dimensional combinatorial optimization problem to a one-dimensional optimization problem, which then leads to the convergence to optimal social welfare under certain conditions. This, and other results, certainly require a very good understanding of the underlying structure.

Although the writing is quite dense at times, it’s easy to follow the contributions and the reasoning behind the results, how it places in the literature and the challenges/limitations, two of which are clearly stated in the conclusion.

Weaknesses:

What happens in the computational front? There are two points with potential computational hurdles. The first is in the optimization problem in corollary 3.3, where things have to be restricted to specific subclasses of matroids for things to potentially be computed efficiently. However, I think that’s fine since some of the most common matroids have poly-time separation oracles. Do you happen to know how it performs and scales in practice (there can be problems in practice with the ellipsoid method..)? The second point comes from the fact the optimization problem computes a fractional allocation, and then we still have to go back to an integer one. How do we do that? Is proposition 2.4 merely for existence, or we know that we can easily obtain these $p_i$’s for drawing the integral allocation?
Experiments would help address some of the practical aspects and computational concerns, but I understand that the paper is already quite heavy in its theoretical contribution.

I think most of the motivating problems mentioned in the intro (and other well-known ones) are integer allocation problems, and not necessarily always for large markets? I’m not sure that we will be getting always very good approximations efficiently, when calculating the fractional allocations and moving to the integer ones. Also, in general, the deviation ex-post is not great when the number of groups is large, which can be, depending on the application. Do you have any example where the ex-post deviation is (almost) matched? So that we understand that ex-post we can’t do much better than this bound. Also, is it always immediate in large markets, that each group has large enough rank, so that the min is large as the authors state? It’s also that $\min r_c$ is also quite big compared to $C$, for the constraint violations to be small.

This is not exactly a weakness, but the justification of opportunity fairness could be justified a bit better. An important part of believing in the strengths of the paper is believing in the definition of opportunity fairness, since the results and techniques are tailored to it. I appreciate the comparison with proportional fairness and other fairness notions in the appendix, but I don’t know if I’m entirely convinced yet that the potential presence of irrelevant agents is enough justification by itself for disregarding the other fairness notions? Why not make some assumption of the sort that there are no irrelevant agents, i.e., any agent in the problem can be allocated at least some small amount of resources? To me, this (or some other assumption on irrelevant agents) doesn’t sound much different in nature to the large markets assumption, or that there is no dominant group for the result in the semi-random setting.

---

> ### Author Rebuttal · Authors · 2025-07-30
>
> Dear reviewer, thank you very much for your thoughtful review, relevant comments and appreciation of our results!
>
>
> ### **1. Computational Complexity**
>
> We agree that the ellipsoid method tends to be quite slow in practice. However, for many common matroid classes, including transversal and graphic matroids, the fair optimization problem admits a polynomial-size LP formulation, which allows us to use more efficient algorithms such as Simplex or interior point methods.
>
> Indeed, this relates to the concept of extension complexity, which basically refers to the minimal number of constraints needed to describe the independence polytope, possibly adding some variables. Notably, Aprile & Fiorini in their paper `Regular matroids have polynomial extension complexity’ (2019) show that this approach is possible when dealing with regular matroids (which includes all of our examples).
>
> Currently, Proposition 2.4 indeed only establishes the existence of a decomposition into integral solutions. However, because the extreme points of the polytope are integral, this decomposition can be computed efficiently using an algorithmic version of Carathéodory's theorem, see for example [Theorem 6.5.11, Geometric Algorithms and Combinatorial Optimization, Grötschel-Lovász-Schrijver, 1993].
>
> We will include these remarks in Appendix D, with cross-references in the main text.
>
> ### **2. Fractional vs. Integral Allocations**
>
> In multiple fair resource allocation works, randomized (i.e., fractional) allocations are interpreted as satisfying fairness ex-ante, with the implicit understanding that the actual allocation is randomized. We highlight this interpretation in line 170, which we will make clearer, and it does not rely on a large market assumption.
>
> From this perspective, the fractional solution is the optimal ex-ante fair allocation, and Proposition 2.4 offers a method to implement it with ex-post guarantees, which do rely on assumptions like $C/\min_c r(c)$ being small. We will clarify this distinction and the role of each assumption in the final version.
>
> We interpret your question as asking whether Proposition 2.4 provides tight worst-case ex-post guarantees for implementing the optimal ex-ante fair allocation. Let us know if we have misinterpreted your question, we are happy to clarify.  We can show that both the $L_1$ deviation and the fairness loss are tight, up to a constant factor.
>
> - **Worst-case $L_1$ deviation:** Consider a submodular function $f(S) = \max(|S|, C \cdot \mathbf{1}[1 \in S])$, with an optimal fair allocation of $(C/2+1/2,1/2,...,1/2)$. Projecting the Pareto Front over the last $C-1$ coordinates yields a unit cube $[0,1]^{C-1}$, with the optimal fair allocation lying at the cube center. Any integral allocation must be projected to a corner of the cube, and the $\Vert \cdot \Vert_1$ distance between any corner and the center, which is bigger than between the two non-projected points, is $C/2$. Hence, this yields a worst-case lower bound $|X - x|_1 = \Omega(C)$ which is tight up to constants.
>
> - **Worst-case fairness degradation:** Using the construction from Figure 6, suppose group 1 matches up to $r_1$ agents, and the remaining groups share $r_2$ agents, with $(r_1/(C-1),r_2/(C-2),\dots,r_2/(C-2)$ the optimal fair allocation. Let $r_1$ and $r_2$ be such that the nearest integers to $r_1/(C-1)$ and $r_2/(C-2)$ are respectively $r_1/(C-1)-1/2+\epsilon$ and $r_2/(C-2)+1/2-\epsilon$. Hence, for $(C-1)\leq r_2<<r_1$,  we have $$ \frac{(r_1/(C-1)-1/2)/r_1}{(r_2/(C-1)+1/2)/r_2}=\frac{1/(C-1)-1/(2r_1)}{1/(C-1)+1/(2r_2)}= 1- \frac{1/(2r_1)+1/(2r_2)}{1/(C-1)+1/(2r_2)} \approx 1- \frac{C-1}{C-1+2r_2} \leq 1- \frac{C-1}{3r_2}=1-O(\frac{C}{\min_c r(c)}).$$
>
> We agree that $\min_c r(c)$ may not be large in all instances. Our use of the term “large-market” is informal, and additional assumptions (beyond only $\vert E \vert$ large) are needed to ensure small violations. For instance, this is the case of the semi-random setting with the same assumptions as for Proposition 4.7.
>
> ### **3. Fairness Notion Justification**
>
> We do not intend to argue that other fairness notions (e.g., proportional fairness) should be disregarded. Rather, we want to highlight opportunity fairness as a relevant fairness notion in fair resource allocation, with some of its advantages being independence from irrelevant agents, and being motivated by Equality of Opportunity from supervised learning.
>
> The polymatroid characterization in fact extends to the other fairness notions, and studying with additional assumptions the PoF (such as no irrelevant agents, as suggested by the reviewer) does make sense. In fact we even give a brief result in Proposition F.4, showing that in the semi-random setting the PoF of proportional fairness converges to $1$. Nonetheless, we chose to focus on opportunity fairness in this paper to avoid diverting attention across too many distinct fairness notions.
>
> ### **4. Additional Reviewer Questions**
>
> - **Proposition 4.6 (Equal Ranks Assumption):** Yes, this proposition assumes all group ranks are equal. Extending this result seems challenging: the worst-case example that gives PoF $= C - 1$ has independence index $\rho \to 1$, thus the monotonicity of the bound in $\rho$ breaks down when $\max_c r(c)/\min_c r(c)$ becomes large.
>
> - **Proposition 2.5 ($\gamma$-Fairness Relaxation):** The lower bound arises by disregarding the feasibility constraints, and improving the original optimal fair solution by simply scaling non-tight coordinates by $1/\gamma$. The upper bound is obtained via a convex combination between the optimal fair and optimal unfair solution which is made to be exactly $\gamma$-fair, and which remains feasible by convexity. For small $\gamma$, this gives approximately $PoF_\gamma \approx \gamma \cdot PoF$.
>
> - **$C-2$ Bound in Semi-Random Setting:** Indeed, both analyses are tight. While $C - 2$ is better than $C - 1$, this result is negative in spirit: it shows that the presence of a dominant group larger than the rest of all other groups combined can eliminate most of the advantage of the semi-random setting.
>
> - **Section 4.4 Presentation:** We will follow the reviewer's advice and clarify the model assumptions.
>
> We are happy to discuss with the reviewer and answer any potential additional questions during the rebuttal period.

---

> > ### Comment · Reviewer_vFMq · 2025-08-04
> >
> > Thank you for answering all my questions in detail. Overall, I'm positive about the paper. I still have some concerns about its practical implementation (e.g., regarding the time for solving very large instances, the ex-post guarantees, the weighted scenario not being handled, as pointed also by reviewer JYFo). I will keep my initial score.

---

### Decision · Program_Chairs · 2025-09-17

**Decision:**

Accept (poster)

**Comment:**

On the positive side, reviewers appreciated the paper pushing in a new direction and showing that interesting results are possible.  On the negative side, there was concern about the applicability of the model, particularly in light of some of the model assumptions and the results not handling natural extensions such as weights.